# Notch signalling maintains Hedgehog responsiveness via a Gli-dependent mechanism during spinal cord patterning in zebrafish

**Craig T Jacobs, Peng Huang\***

Department of Biochemistry and Molecular Biology, Cumming School of Medicine, Alberta Children's Hospital Research Institute, University of Calgary, Calgary, Canada

**Abstract** Spinal cord patterning is orchestrated by multiple cell signalling pathways. Neural progenitors are maintained by Notch signalling, whereas ventral neural fates are specified by Hedgehog (Hh) signalling. However, how dynamic interactions between Notch and Hh signalling drive the precise pattern formation is still unknown. We applied the PHRESH (PHotoconvertible REporter of Signalling History) technique to analyse cell signalling dynamics in vivo during zebrafish spinal cord development. This approach reveals that Notch and Hh signalling display similar spatiotemporal kinetics throughout spinal cord patterning. Notch signalling functions upstream to control Hh response of neural progenitor cells. Using gain- and loss-of-function tools, we demonstrate that this regulation occurs not at the level of upstream regulators or primary cilia, but rather at the level of Gli transcription factors. Our results indicate that Notch signalling maintains Hh responsiveness of neural progenitors via a Gli-dependent mechanism in the spinal cord.
DOI: https://doi.org/10.7554/eLife.49252.001

**\*For correspondence:**
huangp@ucalgary.ca

**Competing interests:** The authors declare that no competing interests exist.

## Introduction

Patterning of the spinal cord relies on the action of multiple cell signalling pathways with precise spatial and temporal dynamics (*Briscoe and Novitch, 2008*). Neural progenitors in the spinal cord are organised into discrete dorsoventral (DV) domains that can be identified by the combinatorial expression of conserved transcription factors (*Alaynick et al., 2011*; *Dessaud et al., 2008*; *Jessell, 2000*). Differentiated post-mitotic neurons migrate from the medial neural progenitor domain to occupy more lateral regions of the spinal cord.

To achieve precise patterning, the developing spinal cord employs anti-parallel signalling gradients of Bone Morphogenic Protein (BMP) and Hedgehog (Hh) to specify dorsal and ventral cell fates, respectively (*Le Dréau and Martí, 2012*). Cells acquire their fates via sensing both graded inputs. This dual signal interpretation mechanism provides more refined positional information than separate signal interpretation (*Zagorski et al., 2017*). The action of Sonic Hedgehog (Shh) in the ventral spinal cord is one of the most well studied examples of graded morphogen signalling (*Briscoe and Thérond, 2013*; *Cohen et al., 2013*). In vertebrates, Hh signalling requires the integrity of primary cilia, microtubule-based organelles present on the surface of most cells (*Eggenschwiler and Anderson, 2007*). In the absence of the Shh ligand, the transmembrane receptor Patched (Ptc) represses signal transduction by inhibiting the ciliary localisation of the transmembrane protein Smoothened (Smo). When Shh binds to Ptc, Smo inhibition is released, allowing Smo to translocate to the primary cilia (*Corbit et al., 2005*; *Rohatgi et al., 2007*). This leads to the activation of the Gli family of transcription factors, resulting in expression of downstream target genes

such as *ptc*. Shh thus controls the balance between full-length Gli activators and proteolytically processed Gli repressors (*Huangfu and Anderson, 2006*; *Humke et al., 2010*). In mouse, Gli2 is the main activator and its expression does not require active Hh signalling (*Bai et al., 2002*; *Bai and Joyner, 2001*). In zebrafish, Gli1 is the main activator (*Karlstrom et al., 2003*). Although *gli1* is a direct target of Hh signalling, low-level *gli1* expression is maintained in the absence of Hh signalling via an unknown mechanism (*Karlstrom et al., 2003*). It is thought that Hh-independent *gli* expression allows cells to respond to Hh signals. In the ventral spinal cord, it has been shown that both the level and duration of Hh signalling is critical to the correct formation of the discrete neural progenitor domains along the dorsoventral axis (*Dessaud et al., 2010*; *Dessaud et al., 2007*). However, the temporal dynamics of Hh signalling has been challenging to visualize in vivo due to the lack of appropriate tools.

In addition to BMP and Hh signalling, Notch signalling has also been implicated in spinal cord development (*Louvi and Artavanis-Tsakonas, 2006*; *Pierfelice et al., 2011*). In contrast to long-range Hh signalling, the Notch signalling pathway requires direct cell-cell interaction, as both receptor and ligand are membrane bound proteins (*Kopan and Ilagan, 2009*). The Notch receptor, present at the 'receiving' cell membrane, is activated by the Delta and Jagged/Serrate family of ligands, present at the membrane of the neighbouring 'sending' cell. This leads to multiple cleavage events of Notch, the last of which is mediated by a γ-secretase complex that releases the Notch intracellular domain (NICD). NICD then translocates to the nucleus and forms a ternary transcription activation complex with the mastermind-like (MAML) coactivator and the DNA binding protein RBPJ. This activation complex is essential for the transcription of downstream targets, such as the Hes/Hey family of transcription factors (*Artavanis-Tsakonas and Simpson, 1991*; *Pierfelice et al., 2011*). Two major roles of Notch signalling in neural development are to generate binary cell fate decisions through lateral inhibition and to maintain neural progenitor state (*Formosa-Jordan et al., 2013*; *Kageyama et al., 2008*). However, how Notch signalling interacts with Hh signalling during spinal cord patterning is not clear.

During spinal cord patterning, as Hh responsive neural progenitors differentiate, they lose their competence to respond to Hh signals (*Ericson et al., 1996*). This temporal change in Hh responsiveness could be an indirect consequence of neuronal differentiation, or alternatively, an actively regulated process. Recent work in chick suggests the latter scenario. Floor plate induction requires transient high level of Hh signalling followed by termination of Hh responsiveness, which is critical for the proper fate specification (*Ribes et al., 2010*). However, how the temporal gating of Hh responsiveness is controlled remains poorly understood. Using zebrafish lateral floor plate (LFP) development as a model, we previously demonstrated that Notch signalling maintains Hh responsiveness in LFP progenitor cells, while Hh signalling functions to induce cell fate identity (*Huang et al., 2012*). Thus, differentiation of Kolmer-Agduhr" (KA") interneurons from LFP progenitors requires the downregulation of both Notch and Hh signalling. Recent reports provide additional support for cross-talk between these pathways during spinal cord patterning in both chick and mouse (*Kong et al., 2015*; *Stasiulewicz et al., 2015*). Notch activation causes the Shh-independent accumulation of Smo to the primary cilia, whereas Notch inhibition results in ciliary enrichment of Ptc1. Accordingly, activation of Notch signalling enhances the response of neural progenitor cells to Shh, while inactivation of Notch signalling compromises Hh-dependent ventral fate specification. These results suggest that Notch signalling regulates Hh response by modulating the localisation of key Hh pathway components at primary cilia.

Here, we determine the interaction between Notch and Hh signalling during spinal cord patterning in zebrafish. Given the rapid nature of zebrafish development, combined with the long half-lives of fluorescent proteins, conventional GFP reporters do not allow detecting dynamic cell signalling events with high temporal resolution. To circumvent this issue, we recently developed a photoconversion-based PHRESH technique allowing for visualising cell signalling activation and attenuation in live embryos (*Huang et al., 2012*). Using the PHRESH technique, we show that Notch and Hh response display similar spatiotemporal kinetics. Gain- and loss-of function experiments confirm that Notch signalling is required to maintain Hh response in neural progenitors. Surprisingly, Notch signalling doesn't regulate the Hh pathway at the level of Smo or primary cilia, but rather at the level of Gli transcription factors. Together, our data reveal that Notch signalling functions to control the Hh responsiveness of neural progenitors in a primary cilium-independent mechanism.

## Results

### Generation of a Notch signalling reporter

Spinal cord patterning is a dynamic process with complex interactions of cell signalling pathways in both space and time. To visualise the signalling events in a spatiotemporal manner, we have previously developed the PHRESH (PHotoconvertible REporter of Signalling History) technique (*Huang et al., 2012*). This analysis takes advantage of the photoconvertible properties of the Kaede fluorescent protein to visualise the dynamics of cell signalling response at high temporal and spatial resolution. We have utilised the PHRESH technique to visualise Hh signalling dynamics during spinal cord patterning (*Huang et al., 2012*). To apply the same technique to Notch signalling, we generated a reporter line for *her12*, a target gene of Notch signalling (*Bae et al., 2005*). This target was chosen because among other Notch target genes co-expressed with *her12*, such as *her2*, *her4*, and *hes5*, *her12* had the highest level of expression throughout the spinal cord (*Figure 1—figure supplement 1*). By BAC (bacteria artificial chromosome) recombineering, we generated a *her12:Kaede* reporter by replacing the first coding exon of *her12* with the coding sequence for the photoconvertible fluorescent protein Kaede (*Figure 1A*). The resulting *her12:Kaede* BAC contains 135 kb upstream and 63 kb downstream regulatory sequences. The *her12:Kaede* reporter line faithfully recapitulated endogenous *her12* expression (*Figure 1B*). This reporter also responded to different Notch pathway manipulations (*Figure 1C–E*). The zebrafish *mindbomb* mutant is unable to activate Notch signalling due to an inability to endocytose the Delta ligand (*Itoh et al., 2003*). As expected, the expression of *her12:Kaede* was completely absent in the spinal cord of *mindbomb* mutants (*Figure 1C*). Similarly, inhibition of Notch signalling with the small molecule γ-secretase inhibitor LY-411575 (*Fauq et al., 2007*) completely abolished *her12* expression within 4 hr (*Figure 1D*; *Figure 1—figure supplement 2*). By contrast, ectopic expression of NICD (Notch intracellular domain) using the *hsp:Gal4; UAS:NICD* line (*Scheer and Campos-Ortega, 1999*) resulted in upregulation and expansion of the *her12:Kaede* expression domain (*Figure 1E*). These results demonstrate that *her12: Kaede* is a sensitive reporter for Notch pathway activity in the spinal cord. The combination of small molecule inhibitors and the *her12:Kaede* reporter allows us to manipulate and monitor Notch signalling dynamics in a tightly controlled temporal manner.

### Notch and Hh signalling display similar dynamics during spinal cord patterning

Using the *her12:Kaede* reporter of Notch response (*Figure 1*) in parallel with the previously described *ptc2:Kaede* reporter of Hh response (*Huang et al., 2012*), we can observe the timing and duration of both pathway activities in vivo (*Figure 2A*). All responding cells are initially labelled by green-fluorescent Kaede (Kaede$^{green}$), which can be photoconverted to red-fluorescent Kaede (Kaede$^{red}$) at any specific time ($t_0$). If the cell has finished its signalling response prior to $t_0$, only perduring Kaede$^{red}$ will be detected. Conversely, if the cell begins its response after $t_0$, only newly synthesised, unconverted Kaede$^{green}$ will be present. Finally, if the cell continuously responds to the signalling both before and after $t_0$, a combination of newly-synthesised Kaede$^{green}$ and perduring Kaede$^{red}$ can be observed and the cell will appear yellow. Thus, Kaede$^{red}$ represents 'past response' before $t_0$, Kaede$^{green}$ indicates 'new response' after $t_0$, whereas Kaede$^{red+green}$ corresponds to 'continued response' through $t_0$ (*Figure 2A*; *Video 1*). For example, if the embryo is photoconverted at 36 hpf (hours post-fertilization) and imaged 6 hr post-conversion at 42 hpf (36 hpf + 6 hr), Kaede$^{red}$ cells have terminated their signalling response before 36 hpf, Kaede$^{green}$ cells initiate the signalling response between 36 and 42 hpf, while Kaede$^{red+green}$ cells have sustained signalling response from before 36 hpf and up to a point before 42 hpf. In our experiments, we photoconverted both *ptc2: Kaede* and *her12:Kaede* embryos at 6 hr intervals throughout spinal cord development, and imaged their Kaede fluorescent profiles 6 hr post-conversion of each time point. The time interval of 6 hr was chosen as it allowed time for higher levels of Kaede$^{green}$ to be synthesised while still providing high temporal resolution. We used 3D reconstruction of lateral z-stacks to generate transverse views in order to analyse both the dorsoventral and mediolateral signalling profiles at each time point. Through quantifying Kaede$^{green}$ fluorescence intensity along the dorsoventral (DV) and mediolateral (ML) axes at multiple points along the anterior-posterior axis, we generated representative signalling profiles at each stage to further visualise and compare the spatial dynamics of active signalling

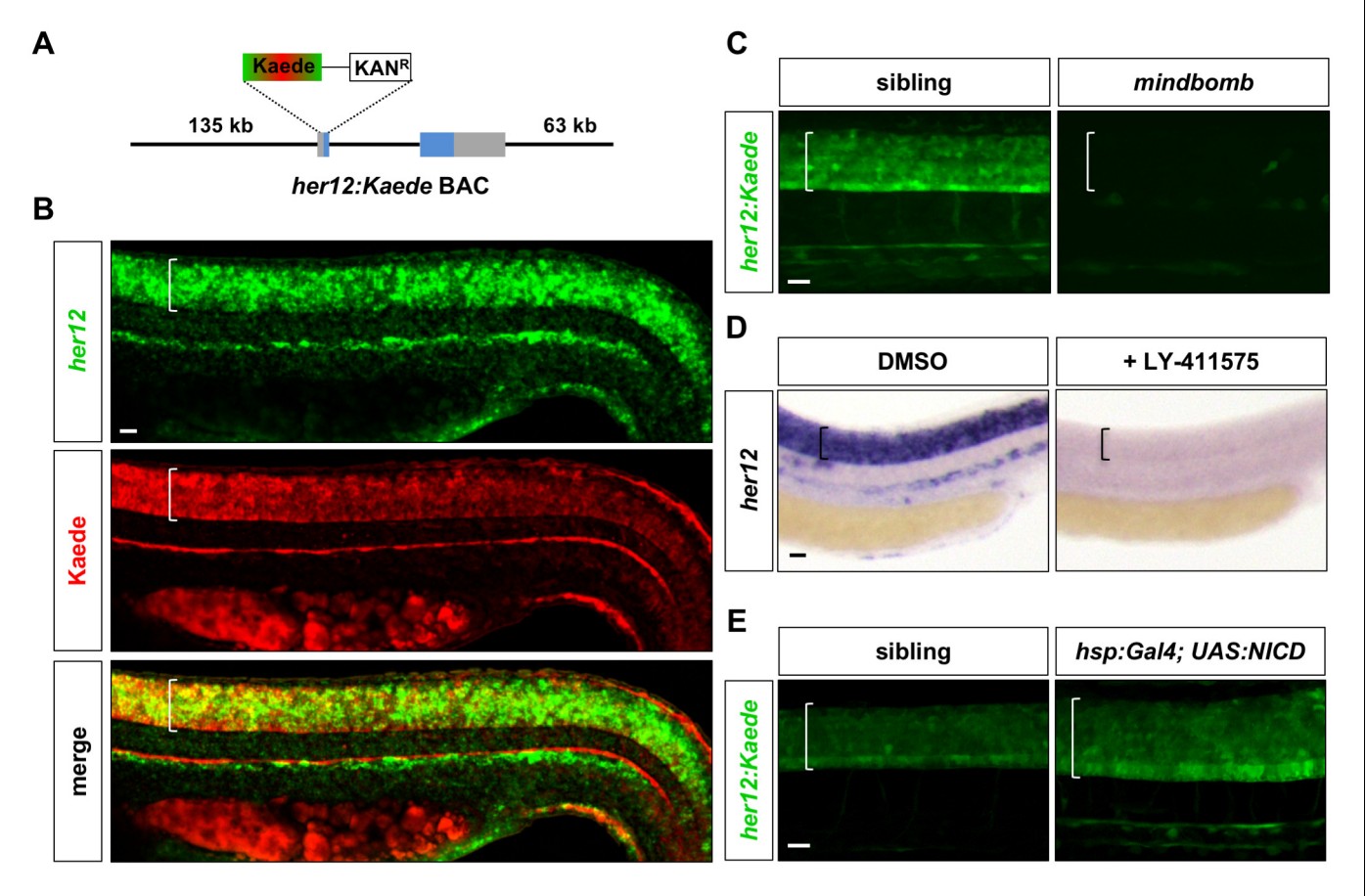

**Figure 1.** Characterization of the *her12:Kaede* reporter. (**A**) Schematic drawing of the *her12:Kaede* BAC reporter. A BAC containing the *her12* locus and surrounding regulatory elements was modified to replace the first exon of *her12* with a cassette containing the coding sequence of Kaede and a Kanamycin resistance gene. *her12* coding exons are highlighted in blue. (**B**) *her12:Kaede* expression, shown by immunohistochemistry using the Kaede antibody (red), recapitulated endogenous *her12* expression, shown by fluorescent in situ hybridisation using the *her12* probe (green). *n* = 8 embryos. (**C**) *her12:Kaede* expression was completely lost in *mindbomb* mutants at 36 hpf. *n* = 6 embryos per genotype. (**D**) Inhibition of Notch signalling by LY-411575 from 20 to 30 hpf completely abolished *her12* expression compared to DMSO treated controls. *n* = 20 embryos per staining. (**E**) Activation of Notch signalling by *hsp:Gal4; UAS:NICD* at 13 hpf resulted in expanded and increased *her12:Kaede* expression at 27 hpf when compared to sibling controls. *n* = 4 embryos per genotype. Brackets in B-E denote the extent of the spinal cord. Scale bars: 20 μm.

DOI: https://doi.org/10.7554/eLife.49252.002

The following figure supplements are available for figure 1:

**Figure supplement 1.** Co-expression of *her12* and other Notch target genes in the spinal cord.

DOI: https://doi.org/10.7554/eLife.49252.003

**Figure supplement 2.** LY-411575 and cyclopamine are efficient inhibitors of Notch and Hh signalling, respectively.

DOI: https://doi.org/10.7554/eLife.49252.004

response (*Figure 2B–D*, graphs). Importantly, these signalling profiles were largely similar throughout the anterior-posterior axis of the photoconverted region (*Videos 2* and *3*) and between different embryos (*Figure 2—figure supplement 1*). Through changing the timing of photoconversion, we were able to create a comprehensive spatiotemporal map of cell signalling dynamics in live embryos (*Figure 2B–D*; *Figure 2—figure supplement 1*; *Figure 2—figure supplement 2*).

Based on spatiotemporal maps of Notch and Hh response, we divided the signalling dynamics of spinal cord development into three general phases: 'signalling activation' phase from 24 to 42 hpf, 'signalling consolidation' phase from 42 to 66 hpf, and 'signalling termination' phase from 66 to 78 hpf. In the first 'signalling activation' phase, active Notch response occurred along the entire dorso-ventral axis of the spinal cord (*Figure 2B*, right), while active Hh response constituted roughly the ventral 75% of the spinal cord in a graded manner (*Figure 2B*, left). Both pathways had a wide peak

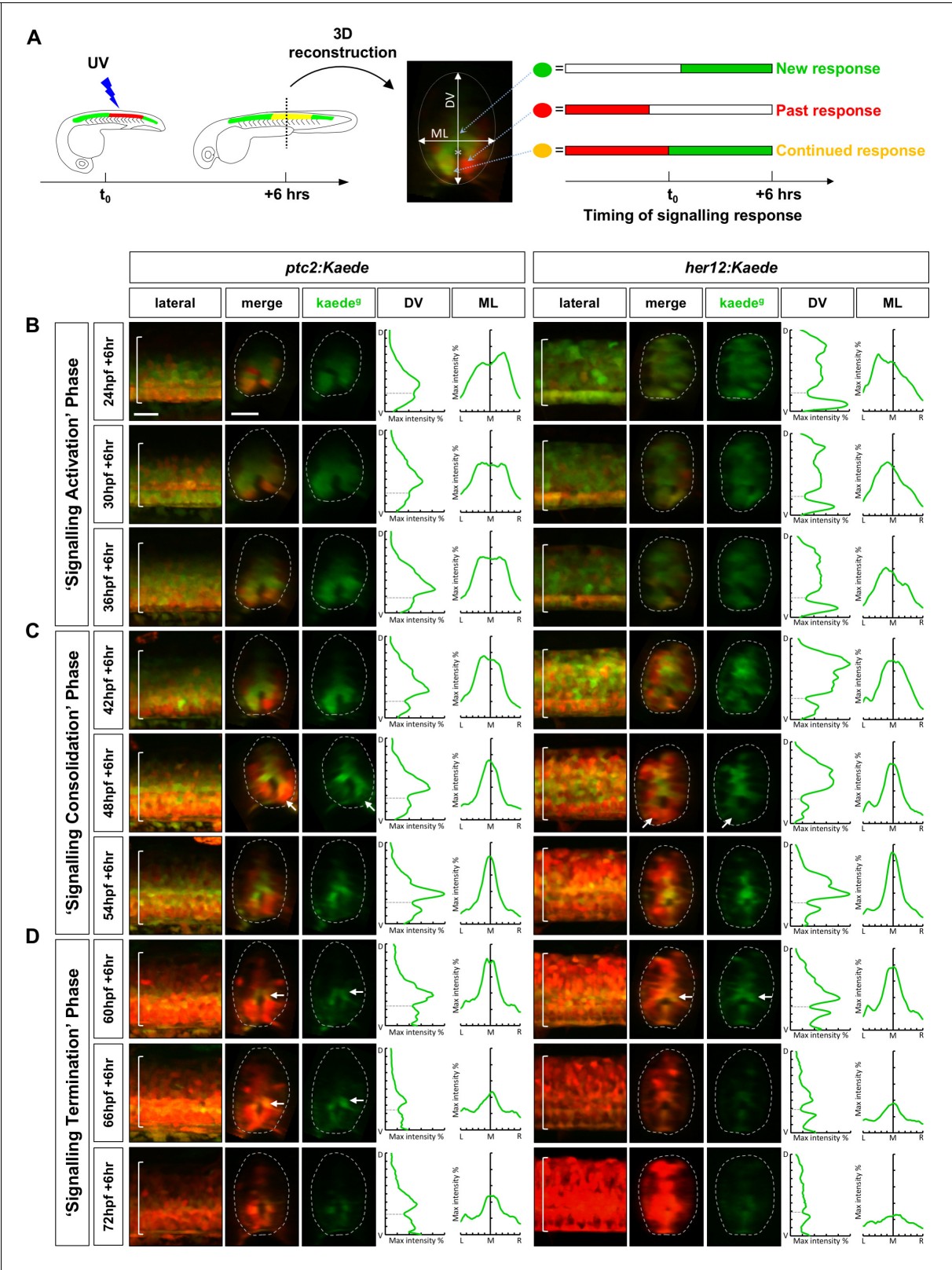

**Figure 2.** PHRESH analysis of the temporal dynamics of Notch and Hh signalling. (**A**) Schematic representation of the experimental design. A section of the spinal cord above the yolk extension was photoconverted by the UV light at $t_0$ and the fluorescent profile was analysed after 6 hours. 3D reconstruction of the spinal cord allowed the identification of cells that have either new signalling response after $t_0$ (green), continued response from before and after $t_0$ (yellow), or have ended signalling response before $t_0$ (red). The graphical signalling profiles were generated from the dorsoventral

*Figure 2 continued on next page*

*Figure 2 continued*

axis (DV) and the mediolateral axis (ML) where indicated. (**B–D**) Time course of Hh and Notch signalling dynamics by PHRESH analysis. *ptc2:Kaede* and *her12:Kaede* embryos were photoconverted at specific time points (t_0, indicated by hpf) and imaged at 6 hr post photoconversion. Lateral views of confocal projections and transverse views of single slices are shown. Kaede[g] panels show de novo synthesised Kaede[green] after t_0, while the merge panels show both previous Kaede[red] expression and new Kaede[green] expression. The graphs show the Kaede[g] fluorescent intensity along the DV and ML axes for each representative embryo. The max intensity axes are 0–50% while the DV/ML axes display the full extent of the transverse section. The dotted lines in the graphs represent the position of the spinal canal. Three distinct phases of signalling response were observed: 'signalling activation' phase between 24 hpf and 42 hpf (**B**); 'signalling consolidation' phase between 42 hpf and 60 hpf (**C**); and 'signalling termination' phase between 60 hpf and 78 hpf (**D**). Arrows in C indicate ventral cells that have terminated response. Arrows in D highlight medial cells right above the spinal canal that remain responsive. Brackets in lateral views and dotted lines in transverse views denote the extent of the spinal cord. *n* = 4 embryos per condition. Scale bars: 20 μm.

DOI: https://doi.org/10.7554/eLife.49252.005

The following figure supplements are available for figure 2:

**Figure supplement 1.** PHRESH signalling profiles show similar trends across multiple embryos.

DOI: https://doi.org/10.7554/eLife.49252.006

**Figure supplement 2.** Notch and Hh response profiles after the 'signalling termination' phase.

DOI: https://doi.org/10.7554/eLife.49252.007

**Figure supplement 3.** *her12* and *ptc2* are expressed in the same domain corresponding to *sox2*[+] neural progenitors.

DOI: https://doi.org/10.7554/eLife.49252.008

of response across the majority of the mediolateral axis, with the weakest response occurring at the lateral edges of the spinal cord. This pattern is consistent with the model that Notch signalling maintains neural progenitor domains, whereas Hh signalling patterns the ventral spinal cord. Interestingly, we found that the signalling response was not entirely homogeneous. In *her12:Kaede* embryos, the majority of cells showed continued Notch response throughout the 'signalling activation' phase, but there were some isolated cells in which Kaede expression was completely absent. In *ptc2:Kaede* embryos, some cells had terminated their Hh response (marked by Kaede[red]), while the majority of cells with the same dorsoventral positioning had continued Hh response. The differential Hh response at the same dorsoventral axis is reminiscent of the differentiation of the lateral floor plate domain (*Huang et al., 2012*).

In the 'signalling consolidation' phase (*Figure 2C*), we observed a dramatic remodelling of the response profiles of both pathways. First, there was an extensive increase in *Kaede[red]* domains for both reporters, indicating the termination of signalling response in these cells. This loss of response was localised to the ventral and lateral regions for Hh signalling (*Figure 2C*, left) and the ventral, lateral and dorsal regions for Notch signalling (*Figure 2C*, right). Second, the number of *Kaede[red]* cells increased as the 'signalling consolidation' phase progressed. Finally, the active signalling domain consolidated into a tight medial region, which sharpened further to encompass 1–2 cell tiers directly dorsal to the spinal canal corresponding to the sharp peaks in DV and ML signalling profiles (*Figure 2C*, 54 hpf + 6 hr).

Finally, during the 'signalling termination' phase (*Figure 2D*), both active Notch and active Hh responses (Kaede[green]) slowly reduced to the basal level, and most of the spinal cord was marked by Kaede[red] by the end of this phase. The active Notch response was notably weaker and restricted to a small medial domain above the spinal canal by 66 hpf before returning to a basal level by 72 hpf. Active Hh response

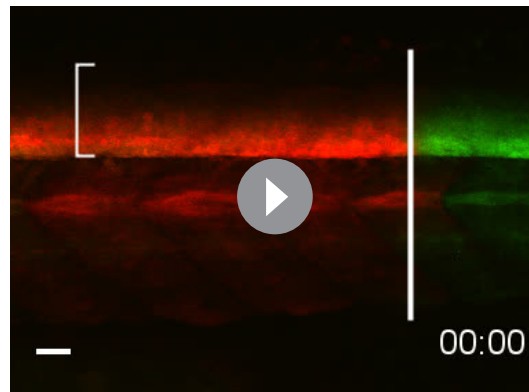

**Video 1.** The re-emergence of *ptc2:Kaede* expression after photoconversion. A *ptc2:Kaede* embryo was photoconverted at 28 hpf and then underwent time-lapse imaging for 18 hr. The vertical line indicates the boundary between photoconverted and unconverted regions at the start of the movie. Bracket indicates the extent of the spinal cord. *n* = 2 embryos. Scale bar: 20 μm.

DOI: https://doi.org/10.7554/eLife.49252.009

**Video 2.** The *ptc2:Kaede* response profile along the anterior-posterior axis. *ptc2:Kaede* embryos were photoconverted at 48 hpf and imaged 6 hr after. Individual transverse sections generated by 3D reconstruction were prepared into a video. The first frame is the most anterior slice and each subsequent frame moves further posterior through the embryo. The merge (left) and *Kaede^green* (right) channels are shown. The spinal cord is denoted by solid lines and the active signalling domain (*Kaede^green*) above the spinal canal is indicated by an arrowhead. Note that *Figure 2C* shows one single slice in the middle of the converted region. *n* = 4 embryos. Scale bar: 20 μm.
DOI: https://doi.org/10.7554/eLife.49252.010

**Video 3.** The *her12:Kaede* response profile along the anterior-posterior axis. *her12:Kaede* embryos were photoconverted at 48 hpf and imaged 6 hr after. Individual transverse sections generated by 3D reconstruction were prepared into a video. The first frame is the most anterior slice and each subsequent frame moves further posterior through the embryo. The merge (left) and *Kaede^green* (right) channels are shown. The spinal cord is denoted by solid lines and the active signalling domain (*Kaede^green*) above the spinal canal is indicated by an arrowhead. Note that *Figure 2C* shows one single slice in the middle of the converted region. *n* = 4 embryos. Scale bar: 20 μm.
DOI: https://doi.org/10.7554/eLife.49252.011

displayed similar, albeit slightly delayed, kinetics compared to the Notch response. In *ptc2:Kaede* embryos, the *Kaede^green* signal marked a small medial domain at 66 hpf and 72 hpf, and reduced to the basal level by 78 hpf. After the end of the 'signalling termination' phase, both pathways remained at the basal level as spinal cord development progressed (*Figure 2—figure supplement 2*).

Comparison of spatiotemporal signalling profiles reveals that Hh and Notch signalling share similar responsive domains. To examine this directly in the same embryo, we performed double fluorescent in situ hybridisation to visualise *her12* and *ptc2* expression together during all three signalling phases of spinal cord development (*Figure 2—figure supplement 3A*). During the 'signalling activation' phase (*Figure 2—figure supplements 3A*, 24 hpf), *ptc2* expression constituted the ventral portion of the *her12* expression domain, while during the 'signalling consolidation' and 'signalling termination' phases (*Figure 2—figure supplements 3A*, 48 hpf and 72 hpf, respectively), *ptc2* and *her12* expression was present within the same restricted medial domain. These results confirm that Notch and Hh response are active in the same cells of the spinal cord. Indeed, double labelling with neural progenitor cell marker *sox2* showed that the medial domain with continued Notch and Hh response corresponded to the *sox2^+* neural progenitor domain (*Figure 2—figure supplement 3B–C*).

Together, our PHRESH analysis reveals that Hh signalling response follows similar spatiotemporal kinetics as Notch signalling response during spinal cord patterning, raising the possibility that there is a functional relationship between these two signalling pathways.

## Notch signalling maintains Hh response

To explore the interaction between the Notch and Hh signalling pathways, we first performed loss-of-function experiments combining small molecule inhibitors with PHRESH analysis (*Figure 3A*). We used the Smo antagonist cyclopamine (*Chen et al., 2002*) and the γ-secretase inhibitor LY-411575 to block Hh and Notch signalling, respectively, in a temporally controlled manner. Cyclopamine significantly reduced *ptc2* expression within 4 hr, whereas LY-411575 dramatically downregulated *her12* expression within 2 hr, and completely abolished it after 4 hr of incubation (*Figure 1—figure supplement 2*). To ensure complete inhibition of signalling response by the point of photoconversion, *ptc2:Kaede* and *her12:Kaede* embryos were incubated with the inhibitors starting from 20 hpf, photoconverted at 24 hpf and then imaged at 30 hpf, comprising 10 hr of total drug inhibition. As expected, cyclopamine treated *ptc2:Kaede* embryos displayed a marked reduction in the amount of de novo synthesised *Kaede^green* compared to controls at 30 hpf (*Figure 3A*). Similarly, LY-411575 treated *her12:Kaede* embryos showed an almost complete loss of *Kaede^green*. In the reciprocal experiments, when *her12:Kaede* embryos were treated with cyclopamine, there was little effect on the levels of *Kaede^green*. However, LY-411575 treated *ptc2:Kaede* embryos showed a dramatic

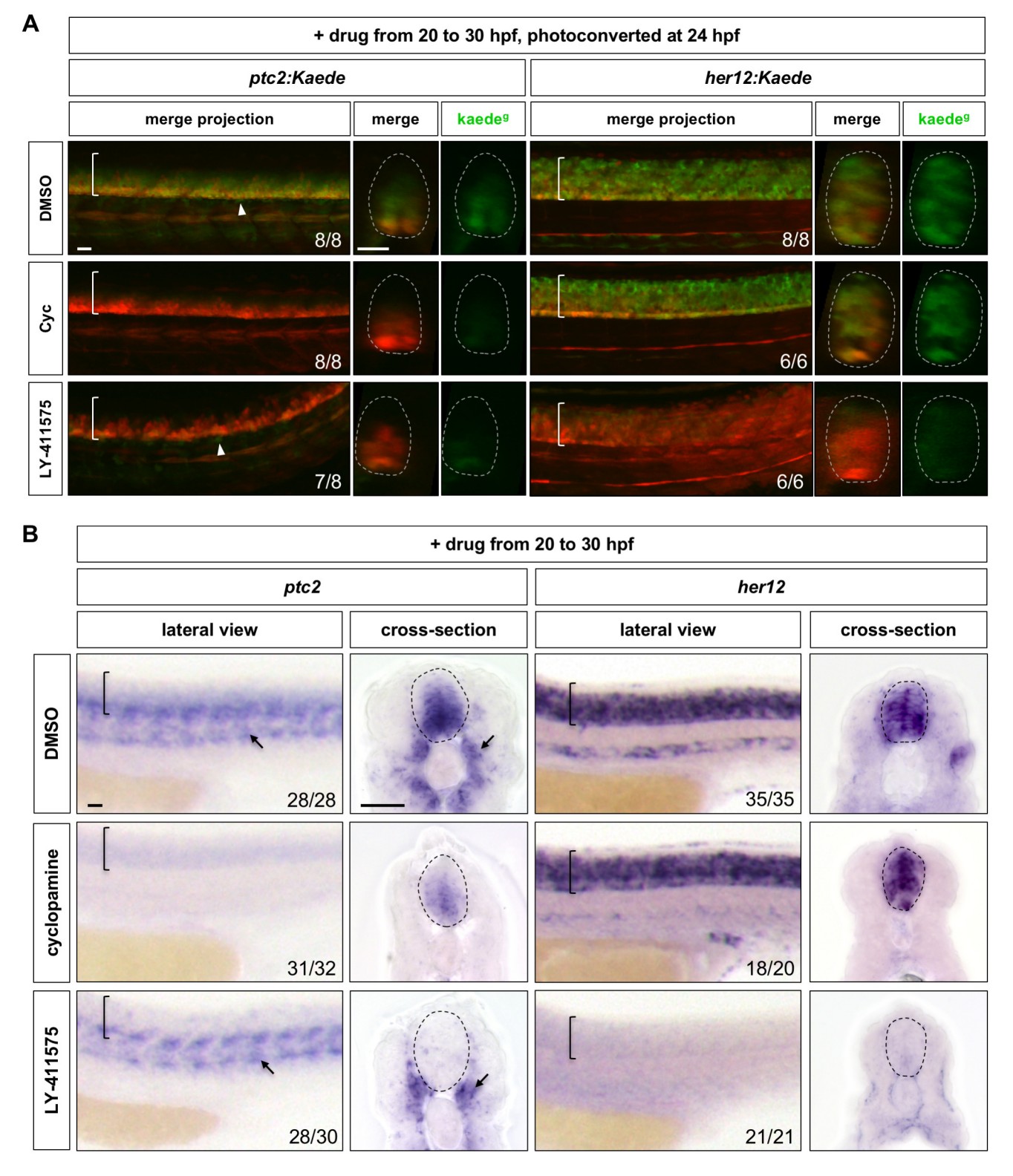

**Figure 3.** Inhibition of Notch signalling abolishes Hh response in the spinal cord. (**A**) *ptc2:Kaede* and *her12:Kaede* embryos were incubated with DMSO, cyclopamine (Cyc) or LY-411575 from 20 to 30 hpf, photoconverted at 24 hpf and imaged at 30 hpf. Lateral views of confocal projections and transverse views of single slices are shown. Kaede[g] panels show de novo synthesised Kaede[green], while the merge panels show both previous Kaede[red] expression and new Kaede[green] expression. Arrowheads highlight *Kaede[green]* cells with active Hh response surrounding the notochord. (**B**) Wild-type

*Figure 3 continued*

embryos were treated with DMSO, cyclopamine, or LY-411575 from 20 to 30 hpf, and stained with *ptc2* or *her12*. Arrows indicate *ptc2* expression in somites. Brackets in lateral views and dotted lines in transverse views in A and B denote the extent of the spinal cord. The *n* number for each staining is shown. Scale bars: 20 μm.

DOI: https://doi.org/10.7554/eLife.49252.012

The following figure supplement is available for figure 3:

**Figure supplement 1.** Inhibition of Notch signalling results in loss of Hh response in the spinal cord.

DOI: https://doi.org/10.7554/eLife.49252.013

reduction in the levels of Kaede^green, reminiscent of cyclopamine treated *ptc2:Kaede* embryos (*Figure 3A*). These results suggest that Notch signalling is required for maintaining Hh response, but not vice versa. Interestingly, despite the inhibition of Notch signalling, cells outside of the spinal cord in *ptc2:Kaede* embryos maintained their normal Hh response, indicated by Kaede^green expression (Arrowheads in *Figure 3A*). This result suggests that regulation of Hh response by Notch signalling is tissue specific.

To confirm these observations from our PHRESH analysis, we performed RNA in situ hybridisation following small molecule inhibition (*Figure 3B*). Wild-type embryos were treated with cyclopamine or LY-411575 from 20 to 30 hpf. When Hh signalling was inhibited by cyclopamine, *ptc2* expression in the spinal cord was significantly reduced but not abolished, while *her12* expression in the spinal cord remained unchanged. By contrast, blocking Notch signalling by LY-411575 resulted in complete loss of both *her12* and *ptc2* expression in the spinal cord. As seen in the PHRESH analysis, *ptc2* expression in cells outside of the spinal cord, such as the somites, was largely intact even after Notch inhibition. To confirm the effects of LY-411575 treatment, we examined *ptc2* expression in *mind-bomb* mutants as well as embryos injected with morpholinos targeting both *rbpja* and *rbpjb. rbpja/b* genes (previously known as *Su(H)1* and *Su(H)2*) encode DNA-binding transcription factors required for Notch response (*Echeverri and Oates, 2007*; *Sieger et al., 2003*). In both cases, *ptc2* expression was abolished in the spinal cord but largely unaffected in somites (*Figure 3—figure supplement 1A–B*), resembling the phenotype of LY-411575-treated embryos. Together, these results are consistent with our model that Notch signalling regulates Hh response specifically in the spinal cord. It is also interesting to note that cyclopamine treated embryos showed residual levels of *ptc2* expression in the spinal cord, whereas LY-411575 treatment completely eliminated *ptc2* expression (*Figure 3B*). It has been shown that zebrafish *smoothened* mutants maintain low-level *gli1* expression in the spinal cord independent of Hh signalling, similar to cyclopamine treated embryos (*Karlstrom et al., 2003*). The complete loss of *ptc2* expression after Notch inhibition suggests that in contrast to cyclopamine, Notch signalling controls Hh response via a different mechanism, likely downstream of Smo.

In converse experiments, we utilised gain-of-function tools to test the interactions between Notch and Hh signalling (*Figure 4*). To activate Hh signalling, we used a transgenic line expressing heat shock inducible TagRFP-tagged rSmoM2, a constitutively activate rat Smo mutant (*hsp:rSmoM2-tRFP*). Induction of rSmoM2 at 11 hpf caused a dramatic upregulation of *ptc2* expression throughout the embryo by 24 hpf, but did not affect *her12* expression in the spinal cord when compared to the control (*Figure 4A*). Interestingly, activation of Hh signalling resulted in a substantial increase in *her12* expression around the dorsal aorta and surrounding vasculature (*Figure 4A*), suggesting that Hh signalling is upstream of Notch response in the vasculature, the opposite relationship to in the spinal cord. To activate ectopic Notch signalling, we induced the expression of the constitutively active Notch intracellular domain (NICD) in *hsp:Gal4; UAS:NICD* double transgenic embryos. Induction of NICD at 11 hpf resulted in a spinal cord specific upregulation and expansion of *ptc2* expression at 24 hpf, while the *ptc2* expression pattern external to the spinal cord was unaffected (*Figure 4B*).

As inhibition of Notch signalling is known to drive premature differentiation of neural progenitor cells, one possibility is that loss of Hh response might be an indirect consequence of neuronal differentiation. To examine this scenario, we compared the timing of loss of Hh response (*ptc2*) versus loss of neural progenitor state (*sox2*) after Notch inhibition. Embryos treated with LY-411575 beginning at 21 hpf for 1, 2 or 3 hr, or with DMSO for the entire timecourse, were analysed for *ptc2* and *sox2* expression (*Figure 5A*). Compared to controls, *ptc2* expression in the spinal cord was drastically reduced after 1 hr of LY-411575 treatment, which further diminished to low level limited to only

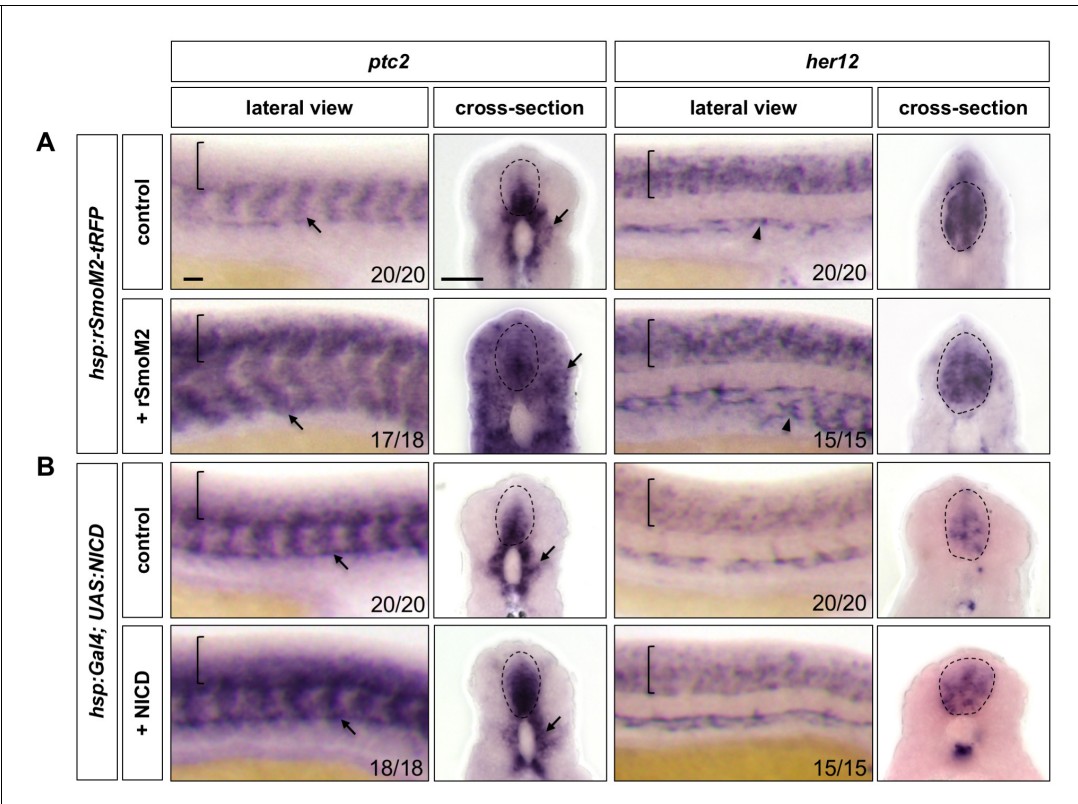

**Figure 4.** Ectopic Notch activation results in increased and expanded Hh response. *hsp:rSmoM2-tRFP* embryos and wild-type controls (**A**), or *hsp:Gal4; UAS:NICD* embryos and wild-type controls (**B**) were heat shocked at 11 hpf and stained for the expression of *ptc2* and *her12* at 24 hpf. Brackets in lateral views and dotted lines in transverse views denote the extent of the spinal cord. Arrows indicate *ptc2* expression in somites. Note that expression of *hsp:rSmoM2-tRFP* resulted in an expansion of *her12* expression in the vasculature compared to control embryos (arrowheads in A). The *n* number for each staining is shown. Scale bars: 20 µm.

DOI: https://doi.org/10.7554/eLife.49252.014

the most ventral region after 3 hr treatment. By contrast, *sox2* expression remained largely unchanged after 1 hr of LY-411575 treatment, and only started to decrease at later time points. Consistent with our observations, when we quantified and compared the normalized expression domain (*Figure 5B*), we found that the *ptc2* domain in the spinal cord was reduced by 44% after 1 hr of LY-411575 treatment, distinct from the 10% decrease for the *sox2* domain. After 2 hr of treatment, both the *ptc2* and *sox2* expression domains were reduced by about 50%, and by 3 hr both expression domains have now shrunk by 61–70% (*Figure 5B*). Since the loss of *ptc2* expression significantly precedes the loss of *sox2* domain, this result suggests that Notch inhibition results in an immediate loss of Hh response, which in turn leads to a loss of neural progenitor state.

Combining our results from the loss- and gain-of-function experiments and the temporal differences between *ptc2* and *sox2* expression following Notch inhibition, we conclude that Notch signalling is required to maintain Hh response, specifically in the spinal cord.

## Notch signalling regulates Hh response downstream of Smo

Our results suggest that Notch signalling regulates Hh response during spinal cord patterning. To explore the molecular mechanisms by which Notch signalling controls Hh response, we determined whether activation of Hh signalling at different points of the pathway can bypass the absence of Notch signalling when the small molecule inhibitor LY-411575 is present (Notch[off] embryos). We first utilised the previously mentioned *hsp:rSmoM2-tRFP* transgenic line to activate Hh signalling at the Smo level (*Figure 6A*). Wild-type control or *hsp:rSmoM2-tRFP* embryos were heat-shocked at 20 hpf, treated with DMSO or LY-411575 for 10 hr, and assayed for gene expression at 30 hpf (*Figure 6B–C*). In DMSO treated embryos, induction of rSmoM2 resulted in substantial expansion of

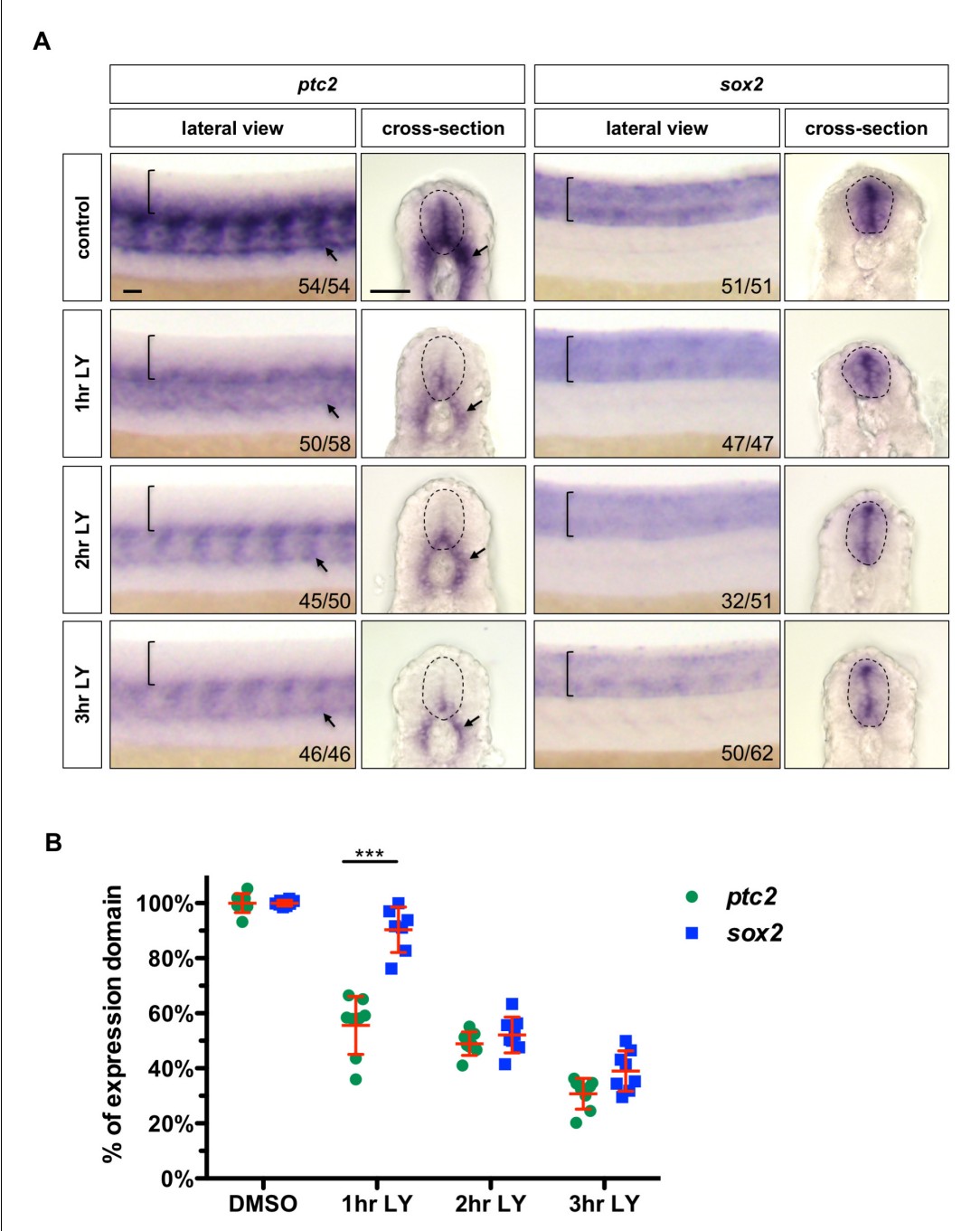

**Figure 5.** Inhibition of Notch signalling results in loss of Hh response followed by loss of neural progenitor identity. (**A**) Wild-type embryos were incubated in LY-411575 from 20 hpf for 1, 2 or 3 hr and DMSO for the entire duration. Embryos were stained for the expression of *ptc2* or *sox2*. Brackets in lateral views and dotted lines in transverse views denote the extent of the spinal cord. Arrows indicate *ptc2* expression in the somites. The *n* number for each staining is shown. Scale bars: 20 µm. (**B**) Transverse sections of embryos were taken from the experiment in A and the extent of the expression domain of *ptc2* and *sox2* was measured and quantified as a percentage of the spinal cord. To directly compare changes in *ptc2* and *sox2* expression domains, the mean of the DMSO treated group was used as the 'control maximum' and all values were normalized as a percentage of their relevant control maximum. Each data point represents the average expression domain percentage of one embryo. *n* = 8 embryos per condition. Data are plotted with mean ± SD. Statistics: Mann-Whitney *U* test. Asterisks representation: p-value<0.001 (***).

DOI: https://doi.org/10.7554/eLife.49252.015

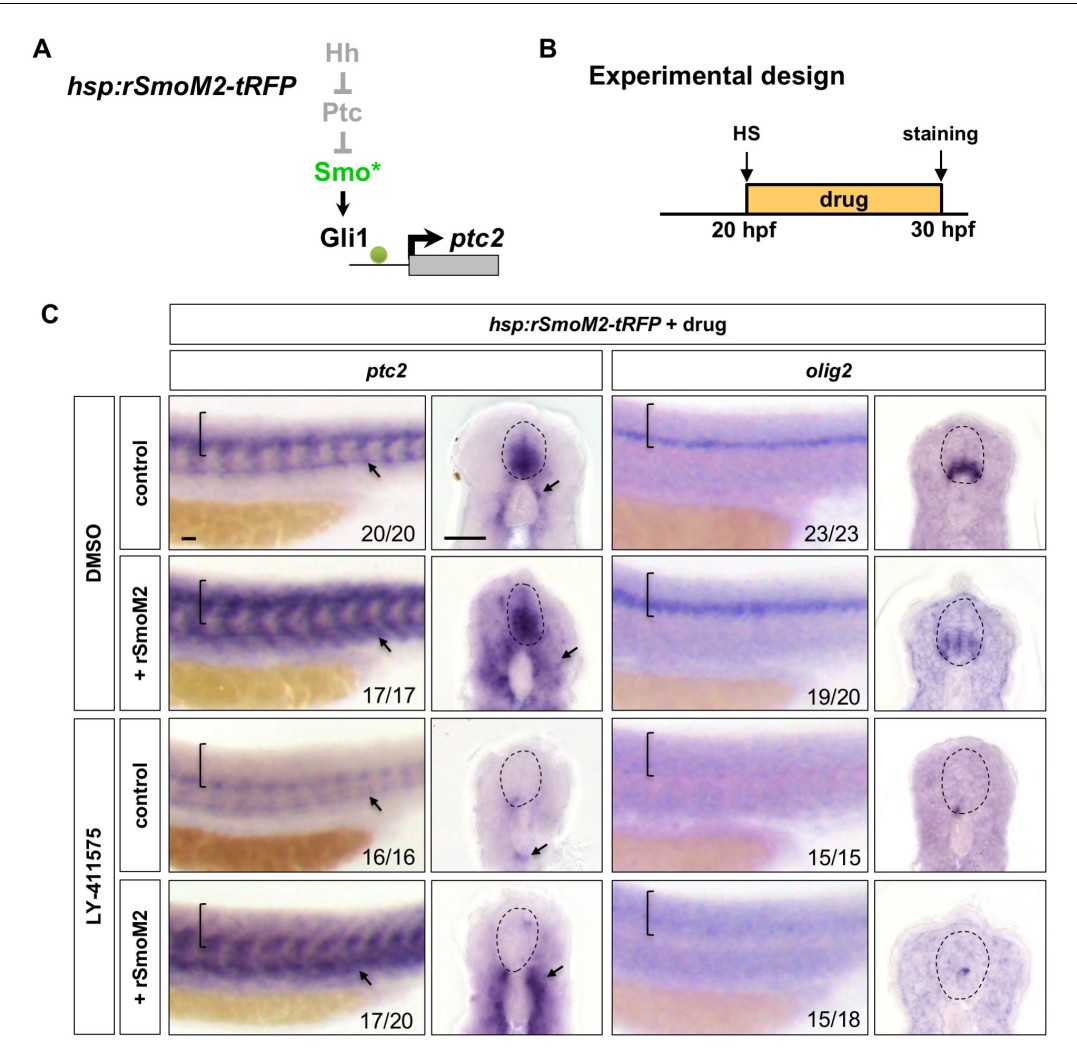

**Figure 6.** Activation of Hh signalling by rSmoM2 cannot rescue Notch[off] spinal cords. (**A**) Schematic representation of the manipulation to the Hh pathway caused by ectopic expression of rSmoM2-tRFP. The point of manipulation is highlighted in green with an asterisk. (**B**) Experimental design in C. (**C**) *hsp:rSmoM2-tRFP* or wild-type control embryos were heat shocked at 20 hpf, and then incubated in either DMSO or LY-411575 until fixation at 30 hpf. Whole mount in situ hybridisation was performed for *ptc2* and *olig2*. Brackets in lateral views and dotted lines in transverse views denote the extent of the spinal cord. Arrows indicate *ptc2* expression in somites. The *n* number for each staining is shown. Scale bars: 20 µm.
DOI: https://doi.org/10.7554/eLife.49252.016

*ptc2* expression in both the somites and the spinal cord (*Figure 6C*). Consistent with this result, activation of Hh signalling by rSmoM2 also led to an expansion of the motor neuron precursor domain, marked by *olig2* expression (*Figure 6C*). By contrast, when Notch signalling was inhibited by LY-411575, induction of both *ptc2* and *olig2* expression by rSmoM2 was still completely blocked in the spinal cord, similar to LY-411575 treated wild-type controls (*Figure 6C*). This result suggests that ectopic activation of Hh signalling at the Smo level is not sufficient to restore Hh response in Notch-[off] spinal cords and further implies that Notch signalling likely regulates Hh response downstream of Smo. Interestingly, induction of rSmoM2 did cause an expansion of *ptc2* expression in the surrounding somites despite Notch inhibition (*Figure 6C*). This observation is consistent with our previous experiments and suggests that this Smo independent mechanism of control is specific to the spinal cord.

# Notch signalling regulates Hh response independent of primary cilia

Vertebrate canonical Hh signalling requires the integrity of primary cilia (*Eggenschwiler and Anderson, 2007*). To test whether Notch signalling feeds into the Hh pathway via primary cilia, we utilised the *iguana* mutant which lacks primary cilia due to a mutation in the centrosomal gene *dzip1* (*Glazer et al., 2010*; *Huang and Schier, 2009*; *Kim et al., 2010*; *Sekimizu et al., 2004*; *Tay et al., 2010*; *Wolff et al., 2004*). In zebrafish, the complete loss of primary cilia, such as in *iguana* mutants, results in reduction of high-level Hh response concomitant with expansion of low-level Hh pathway activity (*Ben et al., 2011*; *Huang and Schier, 2009*). This expanded Hh pathway activation is dependent on low level activation of endogenous Gli1, but does not require upstream regulators of the Hh pathway, such as Shh, Ptc and Smo (*Huang and Schier, 2009*) (*Figure 7A*). Thus, the *iguana* mutant also allows us to determine whether low level activation of the endogenous Gli1 transcription factor is able to restore Hh response in Notch[off] spinal cords. *iguana* mutant embryos or their sibling (heterozygous or wild-type) controls were incubated with DMSO or LY-411575 from 20 hpf and assayed

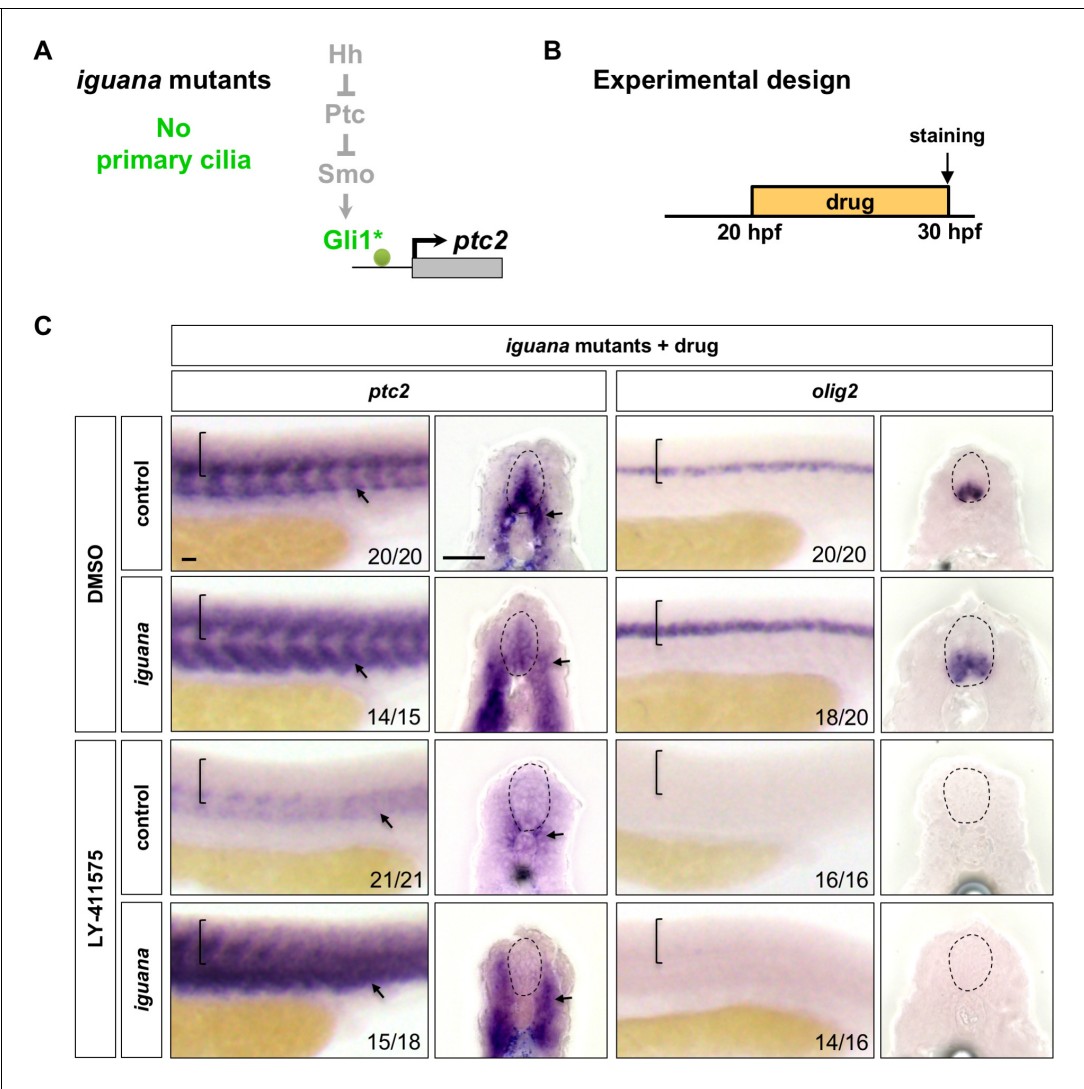

**Figure 7.** Notch signalling regulates Hh response independent of primary cilia. (**A**) Schematic representation of the manipulation to the Hh pathway caused by the loss of primary cilia in *iguana* mutants. The point of manipulation is highlighted in green with an asterisk. (**B**) Experimental design in C. (**C**) *iguana* mutant and sibling control embryos were incubated in either DMSO or LY-411575 at 20 hpf until fixation at 30 hpf. Whole mount in situ hybridisation was performed for *ptc2* and *olig2*. Brackets in lateral views and dotted lines in transverse views denote the extent of the spinal cord. Arrows indicate *ptc2* expression in somites. The *n* number for each staining is shown. Scale bars: 20 μm.
DOI: https://doi.org/10.7554/eLife.49252.017

for gene expression at 30 hpf (*Figure 7B–C*). As shown previously (*Huang and Schier, 2009*), DMSO treated *iguana* mutants showed a reduction of the highest level of *ptc2* expression in the ventral spinal cord, but displayed an overall expansion of the *ptc2* expression domain in both the spinal cord and somites (*Figure 7C*). The low level Hh pathway activation in *iguana* mutants was sufficient to induce and expand the *olig2* domain. Remarkably, we found that Hh pathway activation in *iguana* mutants was completely blocked by Notch inhibition (*Figure 7C*). When *iguana* mutants were treated with LY-411575 at 20 hpf for 10 hr, *ptc2* expression in the spinal cord was completely abolished at 30 hpf, similar to LY-411575 treated sibling controls (*Figure 7C*). This is in contrast with the somites where *ptc2* expression remained expanded in LY-411575 treated *iguana* mutants similar to DMSO treated *iguana* mutants. Consistent with the loss of *ptc2* expression in the spinal cord, LY-411575 treated *iguana* mutants showed almost no *olig2* expression with only rare scattered *olig2* expressing cells (*Figure 7C*). Combined with observations from rSmoM2 experiments, these results suggest that Notch signalling likely functions, in a tissue-specific manner, downstream of Smoothened and the primary cilium in its control of Hh response.

## Ectopic expression of Gli1 partially rescues Hh response in Notch[off] spinal cords

Since low level constitutive activation of endogenous Gli1 in *iguana* mutants is not sufficient to restore Hh response in Notch[off] spinal cords, we hypothesised that Notch signalling regulates Hh response by maintaining *gli1* expression. To test this possibility, we treated wild-type embryos with DMSO, cyclopamine, or LY-411575 at 20 hpf for 10 hr, then assayed for *gli1* gene expression at 30 hpf (*Figure 8A*). In DMSO treated controls, *gli1* expression was present throughout the ventral spinal cord and in the somites. In cyclopamine treated embryos, *gli1* expression was dramatically reduced, but a low level remained in the spinal cord, corresponding to Hh-independent *gli1* transcription (*Karlstrom et al., 2003*). By contrast, LY-411575 treatment completely abolished *gli1* expression in the spinal cord. Strikingly, *gli1* expression in surrounding tissues remained largely unaffected. These results suggest that Notch signalling is required to maintain Hh-independent *gli1* expression in the spinal cord. Interestingly, similar experiments demonstrated that expression of other members of the *gli* genes, *gli2a*, *gli2b* and *gli3*, in the spinal cord, was also largely abolished by LY-411575 treatment (*Figure 8—figure supplement 1*), suggesting that Notch signalling controls the expression of all Gli transcription factors.

We next examined whether overexpression of ectopic Gli1 was sufficient to rescue Hh response in Notch[off] spinal cords (*Figure 8B*). We used an EGFP-Gli1 transgene under the control of a heat shock inducible promoter (*hsp:EGFP-Gli1*) (*Huang and Schier, 2009*). Similar to previous experiments, wild type control or *hsp:EGFP-Gli1* embryos were heat shocked at 20 hpf, treated with DMSO or LY-411575 for 10 hr, and assayed for gene expression at 30 hpf (*Figure 8C*). Induction of ectopic EGFP-Gli1 resulted in the *ptc2* expression domain expanding further dorsally and throughout the spinal cord and the *olig2* domain was also 25% larger than controls (*Figure 8D–E*), a similar phenotype to rSmoM2 induction in DMSO treated embryos. Strikingly, when EGFP-Gli1 induction was followed by LY-411575 treatment, we observed significant *ptc2* expression in the spinal cord, although at a slightly lower level compared to DMSO treated *hsp:EGFP-Gli1* embryos (*Figure 8E*). Critically, the ectopic Gli1-mediated Hh pathway activation in LY-411575 treated *hsp:EGFP-Gli1* embryos was able to restore *olig2* expression to about 63% of the wild-type level (*Figure 8D–E*). Together, these results suggest that, in the spinal cord, Notch signalling regulates Hh response by modulating the Gli1 transcription factor, as ectopic Gli1 can partially rescue the Hh response in Notch[off] spinal cords. This regulation is partly through transcriptional control of *gli1* expression. However, since the ectopic EGFP-Gli1 was unable to rescue the highest level of Hh response and cannot fully restore *olig2* expression in Notch[off] spinal cords, it is possible that Notch signalling plays additional roles in regulating Gli1 activity at the post-transcriptional level, or alternatively, EGFP-Gli1 expression may not be able to fully specify the *olig2* fate in the absence of additional activity from Gli2a, Gli2b and Gli3.

## Discussion

We provide in vivo evidence for cross-talk between two conserved developmental signalling pathways, Notch and Hh signalling, in the zebrafish spinal cord. Through the PHRESH technique, we

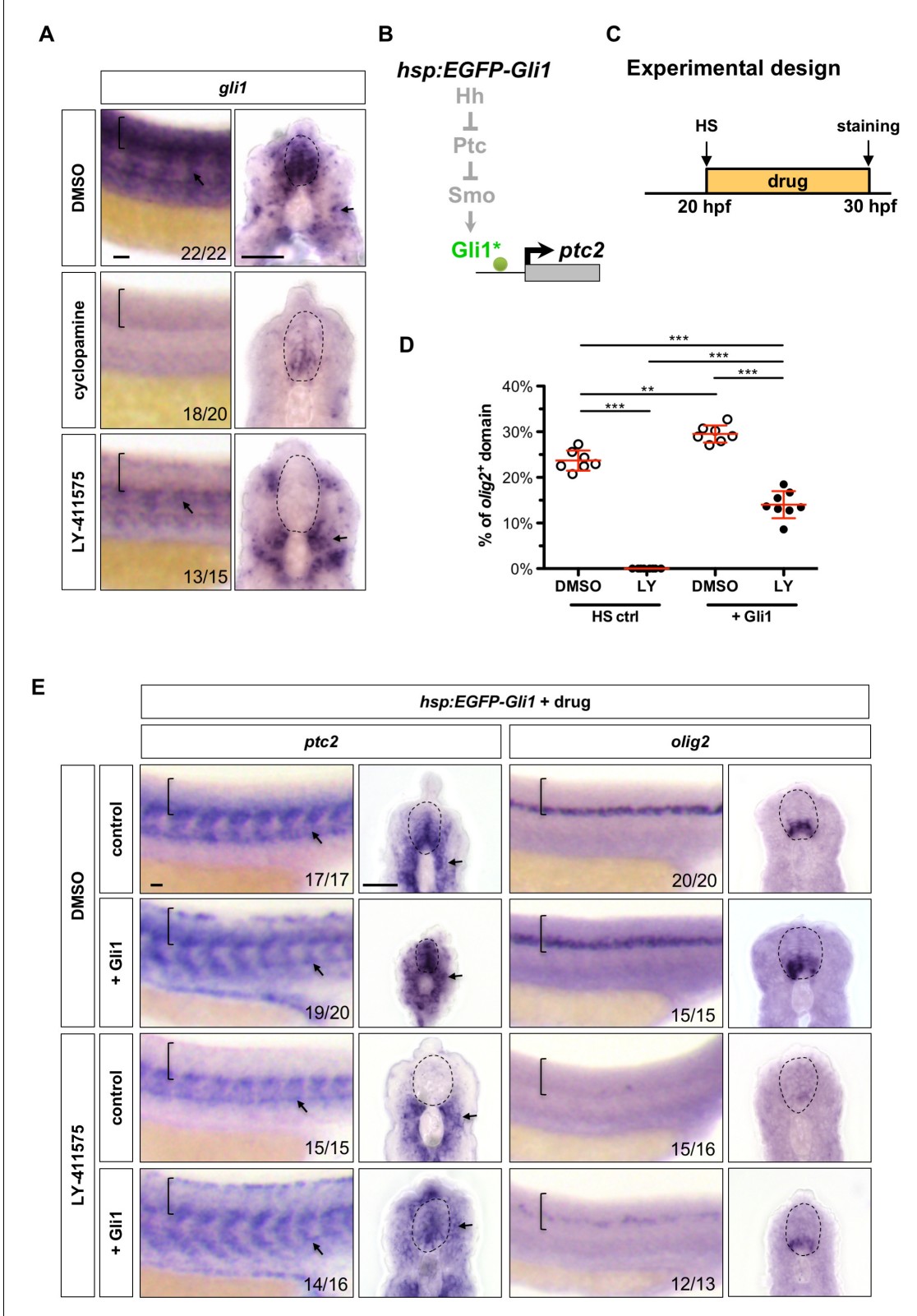

**Figure 8.** Notch signalling regulates Hh response at the Gli level. (A) Whole-mount in situ hybridisation for *gli1* was performed on wild-type embryos treated with DMSO, cyclopamine, or LY-411575 from 20 to 30 hpf. (B) Schematic representation of the manipulation of the Hh pathway caused by ectopic EGFP-Gli1 expression. The point of manipulation is highlighted in green with an asterisk. (C) Experimental design in D-E. (D–E) *hsp:EGFP-Gli1* and wild type control embryos were heat shocked at 20 hpf, and then incubated in either DMSO or LY-411575 until fixation at 30 hpf. Whole mount in

*Figure 8 continued on next page*

*Figure 8 continued*

situ hybridisation was performed for *ptc2* and *olig2*. The extent of the *olig2*[+] expression domain was measured and plotted as a percentage of the spinal cord in D. Each data point represents the average expression domain of one embryo. *n* = 7–8 embryos per condition. Data are plotted with mean ± SD. Statistics: Mann-Whitney *U* test. Asterisks representation: p-value<0.01 (**) and p-value<0.001 (***). Brackets in lateral views and dotted lines in transverse views in A and E denote the extent of the spinal cord. Arrows in A and E indicate *ptc2* expression in somites. The *n* number for each staining is shown in A and E. Scale bars: 20 μm.

DOI: https://doi.org/10.7554/eLife.49252.018

The following figure supplement is available for figure 8:

**Figure supplement 1.** Notch signalling regulates the expression of all Gli family members in the spinal cord.

DOI: https://doi.org/10.7554/eLife.49252.019

observe shared spatiotemporal dynamics of pathway activity throughout spinal cord patterning, highlighting a role for Notch and Hh interaction in neural progenitor maintenance and specification. Using both gain- and loss-of function techniques, we establish a primary cilium-independent mechanism by which Notch signalling permits neural progenitors to respond to Hh signalling via *gli* maintenance.

## Studying cell signalling dynamics using PHRESH

We have previously developed the PHRESH technique to study the dynamics of Hh signalling in vivo (*Huang et al., 2012*). In this study, we demonstrate the versatility of the PHRESH method by correlating the dynamics of Hh and Notch signalling in vivo using the *ptc2:Kaede* reporter and a new *her12:Kaede* reporter. Traditional transcriptional GFP reporters fail to provide temporal information due to GFP perdurance, whereas destabilised fluorescent protein reporters can only provide current activity at the expense of signalling history. By contrast, the PHRESH technique utilises Kaede photoconversion to delineate the cell signalling history in any given time window by comparing newly synthesised Kaede[green] (new signalling) with photoconverted Kaede[red] (past signalling). We envision that PHRESH analysis could be combined with cell transplantation and time-lapse imaging to simultaneously analyse cell lineage and signalling dynamics at single cell resolution. Similar approaches can easily be adapted to study other dynamic events by using photoconvertible fluorescent reporters.

## Spatiotemporal dynamics of Hh and Notch signalling

Using the PHRESH technique, we created a spatiotemporal map of signalling dynamics for the Hh and Notch pathways during spinal cord patterning. Strikingly, Notch and Hh signalling display similar activity profiles. We have characterised these profiles into three general phases: 'signalling activation', 'signalling consolidation', and 'signalling termination'. In the early 'signalling activation' phase, Notch signalling is active throughout the spinal cord, while active Hh response occurs in the ventral ~75% of the spinal cord. During 'signal consolidation', the responsive domain of both pathways sharpens into a small medial domain dorsal to the spinal canal; in 'signalling termination' the response to both pathways returns to a basal level. Our detailed time course reveals three key features of Notch and Hh signalling dynamics. First, early active Hh signalling shows a graded response with the highest level in the ventral domain, as predicted by the classical morphogen model (*Briscoe and Small, 2015*). By contrast, active Notch response does not appear to be graded along the ventral-dorsal axis. Second, despite showing the highest level of Hh response early, the ventral spinal cord terminates Hh response earlier than the more dorsal domains. Therefore, the ventral domain shows higher level Hh response for a shorter duration, whereas the dorsal domain shows lower level response for a longer duration. Our observation is reminiscent of the floor plate induction in chick and mouse embryos, where the specification of the floor plate requires an early high level of Hh signalling and subsequent termination of Hh response (*Ribes et al., 2010*). Our result suggests that Hh signalling dynamics is also evolutionarily conserved. Third, lateral regions of the spinal cord lose both Notch and Hh response before the medial domains. As the active signalling response consolidates into the medial domain, so does the expression of *sox2*, a neural progenitor marker, suggesting that neural differentiation is accompanied by the attenuation of Notch and Hh response. Our observation is consistent with the notion that neural progenitors occupy the medial domain of the spinal cord and that as they differentiate they move laterally.

## Notch signalling regulates Hh response

The loss of Hh response is a necessary step for fate specification, as shown in the chick during floor plate induction (*Ribes et al., 2010*) and in post-mitotic motor neuron precursors (*Ericson et al., 1996*). We have previously shown that the time at which cells attenuate their Hh response is crucial for fate specification in the zebrafish ventral spinal cord (*Huang et al., 2012*). How do neural progenitor cells in the spinal cord maintain their Hh responsiveness until the correct time in order to achieve their specific fates? Multiple lines of evidence indicate that Notch signalling is likely part of this temporal attenuation mechanism controlling Hh responsiveness. First, PHRESH analysis reveals that active Hh response correlates with Notch signalling activity spatially and temporally. The active Hh signalling domain initially constitutes part of the active Notch response domain before following similar kinetics in 'signalling consolidation' and 'signalling termination' phases. This result is consistent with the model that Notch signalling is necessary to maintain Hh responsiveness. Second, loss of Notch signalling either by genetic mutants or by small molecule inhibition results in loss of active Hh response in the spinal cord. In contrast, inhibition of Hh signalling does not affect Notch pathway activity. Critically, upon Notch inhibition, Hh responsiveness in the spinal cord is quickly extinguished prior to the loss of neural progenitor identity, suggesting that the loss of the competence to respond to Hh signals might trigger neuronal differentiation. This idea is supported by the role Notch signalling plays in the transcriptional control of the *gli* genes. In particular, *gli2a* and *gli3* genes are not direct targets of Hh signalling and yet their expression is absent in Notch[off] spinal cords, suggesting a direct role for Notch signalling in controlling the Hh signalling pathway. Indeed, constitutive activation of Notch signalling leads to enhanced Hh pathway activation. Together, our results suggest that Notch signalling functions upstream of Hh signalling in controlling Hh responsiveness during spinal cord patterning.

## Notch signalling gates Hh responsiveness at the level of Gli transcription factors

How does Notch signalling control Hh response? Previous reports have implicated Notch signalling in the regulation of ciliary trafficking of Smo and Ptc (*Kong et al., 2015*; *Stasiulewicz et al., 2015*), thereby modulating cellular responsiveness to Hh signals. However, our previous work in zebrafish has shown that KA" interneuron precursors can turn off their Hh response even when the Hh pathway is constitutively activated by the overexpression of rSmoM2 or by the depletion of *ptc1* and *ptc2* (*Huang et al., 2012*). Similarly, our current work shows that ectopic expression of rSmoM2 is not sufficient to restore Hh response in Notch[off] spinal cords. These results suggest that regulation at the Ptc/Smo level is unlikely the only mechanism that terminates Hh responsiveness. Indeed, we show that in the absence of primary cilia in *iguana* mutants, the low level Hh response remaining due to constitutive Gli1 activation can be completely blocked by Notch inhibition. This result suggests that Notch signalling can regulate Hh response in a primary cilium independent manner, likely at the Gli level. It should be noted that previous studies (*Kong et al., 2015*; *Stasiulewicz et al., 2015*) did not examine the effects of Notch activation and inhibition on the *Gli* genes, so it is plausible that the cilium-dependent mechanism functions in parallel with the cilium-independent mechanism to provide redundant control of Hh responsiveness in neural progenitor cells. Alternatively, since zebrafish ciliary mutants display slightly different effects on Hh response compared to mouse due to differential regulation of *gli* genes (*Huang and Schier, 2009*), it is also possible that this cilium-independent mechanism is specific to zebrafish. In zebrafish, Gli1 functions as the main activator downstream of Hh signalling, although Gli2a, Gli2b and Gli3 also contribute to the activator function (*Karlstrom et al., 2003*; *Ke et al., 2008*; *Tyurina et al., 2005*; *Vanderlaan et al., 2005*; *Wang et al., 2013*). Indeed, inhibition of Notch signalling abolishes both Hh-dependent and Hh-independent *gli1* expression in the spinal cord. Similarly, *gli2a*, *gli2b* and *gli3* expression in the spinal cord is largely eliminated in Notch[off] spinal cords. These results demonstrate that Notch signalling controls the transcription or mRNA stability of all members of the Gli family in the spinal cord. It is possible that *gli* genes are direct targets of Notch signalling, as shown in mouse cortical neural stem cells where N1ICD/RBPJ binding regulates *Gli2* and *Gli3* expression (*Li et al., 2012*). Our model is also consistent with previous work on floor plate induction where the down-regulation of *Gli2* expression has been implicated in the loss of Hh response in mouse and chick floor plate cells (*Ribes et al., 2010*), suggesting that regulation of the transcription of *gli* genes might be an

evolutionarily conserved mechanism to terminate Hh response. Importantly, while ectopic expression of Gli1 from the *hsp:EGFP-Gli1* transgene can re-establish Hh response as indicated by *ptc2* expression in Notch[off] spinal cords, it is unable to fully restore the *olig2* motor neuron precursor domain. This finding suggests two non-mutually exclusive scenarios. The first possibility is that the expression of Gli2a, Gli2b and Gli3 are also required to achieve a full rescue since Notch inhibition abolishes the expression of all *gli* genes in the ventral spinal cord. Alternatively, Notch signalling might play additional roles in regulating Gli1 protein level or activity. A similar mechanism has been suggested in Müller glia of the mouse retina, where Notch signalling controls Gli2 protein levels and therefore Hh response (*Ringuette et al., 2016*). Interestingly, the study by Ringuette et al. favours a translation or protein stability model because Notch manipulation does not alter the *Gli2* transcript level. Together, our work demonstrates that Notch signalling functions to permit neural progenitors to respond to Hh signalling via *gli* transcriptional regulation and potentially Gli protein maintenance. It is conceivable that Notch signalling regulates the Hh pathway at the level of both Ptc/Smo ciliary trafficking and the Gli transcription factors. This dual regulation might ensure efficient termination of Hh response during neuronal differentiation. Intriguingly, the regulation of Hh response by Notch signalling appears to be specific to the neural tissue. The Hh response in the somites is largely unaffected by Notch manipulations, whereas activation of Hh signalling results in an expansion of *her12* expression in the blood vessels, suggesting Notch response is likely downstream of Hh signalling in the vasculature.

In summary, we demonstrate that Notch and Hh signalling share similar spatiotemporal kinetics during spinal cord patterning and that this dynamic interaction is likely required to maintain the neural progenitor zone. We also provide evidence for a primary cilium-independent and Gli-dependent mechanism in which Notch signalling permits these neural progenitors to respond to Hh signalling.

# Materials and methods

**Key resources table**

| Resource type | Designation | Source/Reference | Identifier |
|---|---|---|---|
| Zebrafish strain (*Danio rerio*) | *hsp:Gal4* | *Scheer and Campos-Ortega, 1999*, PMID: 10072782 | RRID:ZFIN_ZDB-ALT-020918-6 |
| Zebrafish strain (*Danio rerio*) | *UAS:NICD* | *Scheer and Campos-Ortega, 1999*, PMID: 10072782 | RRID:ZFIN_ZDB-ALT-020918-8 |
| Zebrafish strain (*Danio rerio*) | *hsp:rSmoM2-tRFP* | This paper. | NA |
| Zebrafish strain (*Danio rerio*) | *hsp:EGFP-Gli1* | *Huang and Schier, 2009* PMID: 19700616 | RRID:ZFIN_ZDB-ALT-110207-11 |
| Zebrafish strain (*Danio rerio*) | *igu[ts294]* (iguana) | *Sekimizu et al., 2004*; *Wolff et al., 2004* PMIDs: 15115751; 15198976 | RRID:ZFIN_ZDB-ALT-980203-1553 |
| Zebrafish strain (*Danio rerio*) | *mib1[ta52b]* (mindbomb) | *Itoh et al., 2003* PMID: 12530964 | RRID:ZFIN_ZDB-ALT-980203-1374 |
| Zebrafish strain (*Danio rerio*) | *her12:Kaede* | This paper: Generated using BAC clone zK5I17 (DanioKey) | NA |
| Zebrafish strain (*Danio rerio*) | *ptc2:Kaede* | *Huang et al., 2012* PMID: 22685423 | RRID:ZFIN_ZDB-ALT-120810-2 |
| Morpholino oligonucleotide | *rbpja/b[MO]* (Previously *Su(H)1+2* MO) | Gene Tools, LLC | ZFIN ID: ZDB-MRPHLNO-070410–11 |
| Antibody | Rabbit polyclonal anti-Kaede | MBL International | Cat# PM012, RRID:AB_592060 |
| Antibody | Goat anti-rabbit IgG, Alexa Fluor 555 | Thermo Fisher Scientific | Cat# A-21428, RRID:AB_2535849 |
| DNA dye | Draq5 | Biostatus | Cat# DR50050, RRID:AB_2314341 |

*Continued on next page*

*Continued*

| Resource type | Designation | Source/Reference | Identifier |
|---|---|---|---|
| Small molecule inhibitor | Cyclopamine | Toronto Chemical | Cat# C988400 |
| Small molecule inhibitor | LY-411575 | Millipore Sigma | Cat# 209984-57-6 |
| Software package | Fiji-ImageJ | *Schindelin et al., 2012* PMID: 22743772 https://fiji.sc | RRID:SCR_002285 |
| Software package | Graphpad Prism | https://www.graphpad.com/scientific-software/prism/ | RRID:SCR_002798 |

## Zebrafish strains

All zebrafish strains used in this study were maintained and raised under standard conditions. All procedures were conducted in accordance with the principles outlined in the current Guidelines of the Canadian Council on Animal Care. All protocols were approved by the Animal Care Committee at the University of Calgary (#AC17-0128). The transgenic strains used were: *her12:Kaede*, *hsp: EGFP-Gli1* (*Huang and Schier, 2009*), *hsp:Gal4* (*Scheer and Campos-Ortega, 1999*), *hsp:rSmoM2-tRFP*, *ptc2:Kaede* (*Huang et al., 2012*), *UAS:NICD* (*Scheer and Campos-Ortega, 1999*). The *hsp: rSmoM2-tRFP* transgenic line was generated by standard Tol2-mediated transgenesis. The *mib1$^{ta52b}$* (*mindbomb*) (*Itoh et al., 2003*) and *igu$^{ts294}$* (*iguana*) (*Sekimizu et al., 2004*; *Wolff et al., 2004*) mutant strains were maintained as heterozygotes, and homozygous embryos were generated by intercrossing heterozygous carriers.

## Generation of the *her12:Kaede* BAC transgenic line

To generate the *her12:Kaede* transgenic line, BAC clone zK5I17 from the DanioKey library that contains the *her12* locus and surrounding regulatory elements was selected for bacteria-mediated homologous recombination following the standard protocol (*Bussmann and Schulte-Merker, 2011*). zK5I17 contains 135 kb upstream and 63 kb downstream regulatory sequences of *her12*. First, an iTol2-amp cassette containing two Tol2 arms in opposite directions flanking an ampicillin resistance gene was recombined into the vector backbone of zK5I17. Next, a cassette containing the Kaede open reading frame and the kanamycin resistance gene was recombined into the zK5I17-iTol2-amp to replace the first exon of the *her12* gene. Successful recombinants were confirmed by PCR analysis. The resulting *her12:Kaede* BAC was co-injected with *tol2* transposase mRNA into wild-type embryos and stable transgenic lines were established through screening for Kaede expression.

## Morpholino injection

To block Notch signalling, morpholino oligonucleotides (Gene Tools, LLC) targeting both *rbpja* (*Su (H)1*) and *rbpjb* (*Su(H)2*) genes (*rbpja/b$^{MO}$*: 5'-CAA ACT TCC CTG TCA CAA CAG G-3') (*Echeverri and Oates, 2007*; *Sieger et al., 2003*) were injected at 0.25 mM into one-cell stage embryos with 1 nl per embryo. Injected embryos were fixed at appropriate stages for in situ analysis.

## In situ hybridisation and immunohistochemistry

All whole-mount in situ hybridisation and immunohistochemistry in this study were performed using standard protocols. We used the following antisense RNA probes: *gli1*, *gli2a*, *gli2b*, *gli3*, *her2*, *her4*, *her12*, *hes5*, *olig2*, *ptc2* and *sox2*. For double fluorescent in situ hybridisation, both dinitrophenyl (DNP) and digoxigenin (DIG) labelled probes were used with homemade FITC and Cy3 tyramide solutions (*Vize et al., 2009*). For immunohistochemistry, rabbit polyclonal antibody to Kaede (1:1000, MBL) was used. The appropriate Alexa Fluor-conjugated secondary antibodies were used (1:500, Thermo Fisher) for fluorescent detection of antibody staining and Draq5 (1:10,000, Biostatus) was used for nuclei staining. All staining was performed in two or more replicates with 15–60 embryos per condition.

## PHRESH analysis

All fluorescent imaging was carried out using the Olympus FV1200 confocal microscope and the Fluoview software. Photoconversion was carried out using the 405 nm laser with a 20x objective. *ptc2:Kaede* and *her12:Kaede* embryos at the appropriate stages were anaesthetised with 0.4% tricaine and then embedded in 0.8% low melting agarose. To achieve complete conversion over a large area, a rectangular area of 1000 by 300 pixels was converted by scanning the area twice with 50% 405 nm laser at 200 µs per pixel. Following confirmation of Kaede[red] expression, embryos were recovered in E3 water with phenylthiourea for 6 hr post-conversion before imaging. Appropriate imaging parameters were established using the unconverted region as a reference to avoid over or under exposure of the Kaede[green] signal. Cross-sections were generated using Fiji-ImageJ software (*Schindelin et al., 2012*) to create a 3D reconstruction of the image, then 'resliced' to yield transverse views of the spinal cord.

To generate PHRESH signalling profiles from the reconstructed transverse views along the dorsoventral axis, three lines were drawn directly through the spinal canal and the fluorescent intensity of Kaede[green] was measured along the lines. Measurements were taken from one section per somite for five neighbouring somites, and the average of the three lines and the five somites was presented in the graph. Similarly, to generate signalling profiles along the mediolateral axis, one line was drawn directly dorsal to the spinal canal and the Kaede[green] fluorescent intensity was analysed in the same manner as described above. In order to control for transgene variability between embryos, the intensity was normalised to the maximum intensity of Kaede[green] in the unconverted region. These profiles were generated using Fiji-ImageJ software then graphically represented using Microsoft Excel.

## Drug treatment

Embryos at the appropriate stage were treated with cyclopamine (Toronto Chemical, 100 µM), LY-411575 (Sigma, 50 µM), or DMSO control in E3 fish water. For PHRESH analysis, embryos were treated from 4 hr prior to the point of conversion until 6 hr post-conversion. To match this, all other drug treatments took place between 20 hpf and 30 hpf except the indicated timecourse experiments.

## Heat shock experiments

To induce expression from the heat shock promoter, embryos at the relevant stage were placed in a 2 ml micro-centrifuge tube in a heat block set to 37°C for 30 min. After heat shock, embryos were transferred back into E3 water in a petri dish and recovered at 28.5°C. For drug treatment after heat shock, embryos were transferred directly from the heat shock to E3 water containing the appropriate drug.

## Cryosectioning

To obtain transverse sections after whole-mount in situ hybridisation, embryos were cryoprotected with 30% sucrose at 4°C before being embedded in OCT compound (VWR) and frozen in the −80°C freezer. Sections were cut between 10–16 µm using a Leica cryostat. Sections were taken from the region of the trunk dorsal to the yolk extension.

## Quantification of expression domains

To quantify the expression domains of *ptc2*, *sox2* and *olig2*, 6–10 cryosections were imaged from each of 7–8 representative embryos. The areas of the expression domain and the corresponding spinal cord were measured using Fiji-ImageJ software. The percentage of the expression domain was calculated by dividing the area of the expression domain by the area of the entire spinal cord. All graphs and statistical analyses were generated using the GraphPad Prism software. For quantifications, standard deviation of the mean was calculated. To analyse significance between two samples, P values were determined by performing the Mann-Whitney *U* test.

## Acknowledgements

We thank the zebrafish community for providing probes and reagents; Holger Knaut for BAC clones; Sarah Childs and members of the Huang laboratory for discussion; and Paul Mains and James McGhee for critical comments on the manuscript.

## Additional information

### Funding

| Funder | Grant reference number | Author |
| --- | --- | --- |
| Natural Sciences and Engineering Research Council of Canada | RGPIN-2015-06343 | Peng Huang |
| Canada Foundation for Innovation | Project 32920 | Peng Huang |
| Alberta Children's Hospital Research Institute | Startup fund | Peng Huang |
| Alberta Children's Hospital Research Institute | Graduate Scholarship | Craig T Jacobs |

The funders had no role in study design, data collection and interpretation, or the decision to submit the work for publication.

### Author contributions

Craig T Jacobs, Peng Huang, Conceptualization, Resources, Data curation, Software, Formal analysis, Supervision, Funding acquisition, Validation, Investigation, Visualization, Methodology, Writing—original draft, Project administration, Writing—review and editing

### Author ORCIDs

Craig T Jacobs (iD) https://orcid.org/0000-0002-0459-2838
Peng Huang (iD) https://orcid.org/0000-0001-7954-8869

### Ethics

Animal experimentation: All procedures was conducted in accordance with the principles outlined in the current Guidelines of the Canadian Council on Animal Care. All protocols were approved by the Animal Care Committee at the University of Calgary (#AC17-0128).

### Decision letter and Author response

Decision letter https://doi.org/10.7554/eLife.49252.022
Author response https://doi.org/10.7554/eLife.49252.023

## Additional files

### Supplementary files

• Transparent reporting form
DOI: https://doi.org/10.7554/eLife.49252.020

### Data availability

All data generated or analysed during this study are included in the manuscript and supporting files.

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
