## [Decision Letter]

[Editors’ note: a previous version of this study was rejected after peer review, but the authors submitted for reconsideration. The first decision letter after peer review is shown below.]

Thank you for submitting your work entitled "Notch signalling maintains Hedgehog responsiveness via a Gli-dependent mechanism during zebrafish spinal cord patterning" for consideration by *eLife*. Your article has been reviewed by a Senior Editor, a Reviewing Editor, and three reviewers. The following individuals involved in review of your submission have agreed to reveal their identity: Bruce Appel (Reviewer #1); Jonathan Eggenschwiler (Reviewer #2).

Our decision has been reached after consultation between the reviewers. Based on these discussions and the individual reviews below, we regret to inform you that your work will not be considered further for publication in *eLife*.

As you can see, there was general consensus that while the initial observations are interesting and potentially important, there would be significant additional work necessary to meet the reviewers' concerns. The main requested revisions are to:

- Provide full quantitation for the data shown;

- Address the important point that Notch inhibition may simply drive cells to differentiate, resulting in the loss of Gli expression and thus loss of competence for a Hh-dependent transcriptional response;

- Discuss discrepancies with previously published results in the literature.

As we feel the additional work needed to address these issues would take more than two months to complete, we are returning your submission to you now in case you wish to submit elsewhere for speedy publication. However, if you address these points and wish to resubmit your work to *eLife*, we would be happy to look at a revised paper. Please note that it would be treated as a new submission with no guarantees of acceptance.

Reviewer #1:

This manuscript addresses the long-standing problem of spinal cord patterning and a recently appreciated, but understudied question of how Notch signaling regulates neural cell responsiveness to Hedgehog signaling.

The main strengths of the paper are:

1) The manuscript is very nicely organized, written and illustrated. It is very easy to read and understand.

2)The work addresses an important problem in neural development and presents new information to the field. Although we have known for a very long time that Notch signaling has a fundamentally important role in maintaining spinal cord progenitors, we still have very little understanding of the details. Recent publications, including from the senior author of this work, have connected Notch signaling to Hh signaling sensitivity. The conclusions drawn by these authors differ substantially from the conclusions of Kong et al., 2015 and Stasiulewicz et al., 2015 and, if valid, would motivate further examination of the mechanistic intersection of Notch and Hh signaling.

3) With the exceptions noted below, the data sufficiently support the conclusions.

The main weaknesses of the paper are:

1) There is no indication of the number of trials performed or embryos analyzed for any experiment. Although most of these manipulations probably produce very clear changes in gene expression and having some sort of statistical data might not be so helpful, I do think it does become important for the experiments portrayed in Figure 7D. These experiments are key to the authors' conclusions and it is essential to know if the images they show represent the data accurately.

2) If Notch signaling sensitizes cells to Hh signaling cell autonomously, as expected if Notch does this by acting on Gli1, then a prediction is that the same cells will express *ptc2* and *her12*. However, the double labeling data of Figure 2—figure supplement 3 do not strongly support that prediction. Do higher magnification, higher resolution images reveal that all, or most, ptc2+ cells also express *her12*?

3) The most important weakness concerns the series of experiments to investigate where Notch acts in the Hh signaling pathway. The authors' approach is to pharmacologically inhibit Notch signaling at 20 hpf and simultaneously manipulate Hh signaling to determine if a particular manipulation can rescue the patterning deficit resulting from loss of Notch function. The problem is that Notch inhibition causes neural progenitors to rather immediately differentiate. Consequently, any subsequent failure in Hh responsiveness might not be due to absence of Notch activity, but because differentiated cells simply are unresponsive to Hh for unrelated reasons. In fact, an interpretation of Figure 2 is that differentiated cells are not Hh signaling active. This is why the data of Figure 7B are so important and therefore must be rock solid. The implication of these results is that elevation of Gli1 can overcome Notch inhibition to maintain some level of Hh signaling and spinal cord patterning. It appears to me that, in this instance, Gli1 overexpression maintains some neural progenitors and prevents their differentiation despite loss of Notch signaling. I therefore predict that delaying heatshock to induce Gli1 expression by an hour would not have the same effect. If that's the case, I'm not convinced that the data can sufficiently support the Notch effect via Gli1 model.

Reviewer #2:

The manuscript entitled "Notch signalling maintains Hedgehog responsiveness via a Gli-dependent mechanism during zebrafish spinal cord patterning" by Jacobs and Huang investigates the relationship between the Notch (N) and Hedgehog (Hh) pathways in patterning the zebrafish neural tube. Specifically, the authors a technique, PHotoconvertible REporter of Signalling History, as well as in situ hybridization, to demonstrate that the N and Hh pathways are active at the same developmental time and in a common population of cells during spinal cord development. The authors use a combination of small molecule inhibitors, genetic mutants, and transgenic (gain-of-function) zebrafish to show that in the spinal neural tube (but generally not in non-neural tissues), Notch pathway activity is required for Hedgehog responses, while Hh pathway is dispensable for N activity. The authors show that Notch pathway activity is required for Hh signaling at a step downstream of Smoothened and cilia. They show that expression of all of the *gli* transcription factors (*gli1, gli2a, gli2b* and *gli3*) relies on N activity and that forced expression of Gli1 is sufficient to partially restore Hh responses when the N pathway has been inhibited. The fact that the rescue is not complete raises the possibility that the Notch pathway may also serve a permissive role for Gli activity at the post-transcriptional levels.

This is important work that advances our understanding of the Hh/N relationship. While previous findings have pointed to a functional relationship, the current work significantly advances our understanding of the mechanistic nature of the relationship and suggests the mechanism acting in zebrafish may differ from that acting in the mouse (Stasiulewicz et al., 2015). The work has been carefully conducted, although it could benefit from some quantitative analysis. Specific comments that relate to overall flow of the writing, data analysis, and interpretation of results are listed below.

Specific comments:

1) To clearly emphasize the importance of the work, it would be useful to emphasize how this work addresses a previously unanswered question in the introduction section. While it is important, with respect to cell fate specification, that the competence of neural cells to respond to Hh signals be shut down at some point in later development, the mechanism by which this competence is restricted is not understood. By demonstrating that N activity is also time-limited and corresponds to the stages when cells are competent for Hh responses) and that Hh responses are dependent on N activity, the authors provide a clue as to the nature of this temporal gating of Hh responsiveness.

2) N values should be clearly stated for each experiment and quantification of data in some of the experiments should be performed.

3) As *gli2a, gli2b* and *gli3* expression in the spinal cord is largely eliminated in Notch(off) spinal cord, is it possible that the failure of hsp:EGFP-Gli1 to fully restore the Olig2+ motor neuron precursor domain is not due to the role of Notch in maintaining Gli1 activity at the post-transcriptional level but, rather hsp:EGFP-Gli1 expression may not be able to fully specify the Olig2 fate in the absence of additional activity from *gli2a, gli2b* and *gli3*. This possibility should be discussed.

Reviewer #3:

The paper by Jacobs and Huang tackles a very interesting problem in the field of signal interpretation. They identify a tissue-specific regulation of Hh signaling that is both surprising and intriguing. They use a range of mutant embryos to uncover the mechanism of this regulation, and identify that this regulation is at the level of Gli itself.

The paper is logically presented and the argument r.e. Hh regulation, constructed sensibly.

However, I have a major concern regarding the imaging and analysis. There is very little quantification of data. We are shown snapshots of "representative" embryos at different timepoints under different conditions, and we are encouraged to take the authors word on the results. For the approach outlined, the authors should be quantifying signal levels and recording how they change in time across multiple embryos. What is the embryo-to-embryo variability? How spatially confined are the results? What is the cell-to-cell variability? The quality of the presented fluorescent images is poor and somewhat unconvincing. There is a lot of information in the PHRESH reporter that they are missing because of out-dated analysis approaches.

A second major issue I have regards the means of perturbation. They rely entirely on drug perturbations to Notch. Are heat shock or Gal4-driven dominant negatives alleles available to test the suppression? The problem with drug treatments is that the whole embryo is affected so deciphering tissue-specific phenotypes needs to be done carefully. Both drug and heat-shock induced perturbations should be used to cross-validate the approach.

To summarise, I think the paper is potentially very interesting and tackles a problem relevant to a broad range of researchers. However, in its current form it appears dated and does not use suitable analysis techniques that give me sufficient confidence in the results.

---

## [Author Response]

[Editors’ note: the author responses to the first round of peer review follow.]

Our decision has been reached after consultation between the reviewers. Based on these discussions and the individual reviews below, we regret to inform you that your work will not be considered further for publication in eLife.As you can see, there was general consensus that while the initial observations are interesting and potentially important, there would be significant additional work necessary to meet the reviewers' concerns. The main requested revisions are to:- Provide full quantitation for the data shown;

We have provided *n* numbers for all experiments throughout the manuscript in figures and figure legends. We have also provided additional quantifications whenever necessary. For PHRESH analysis in Figure 2 and Figure 2—figure supplement 1, we provided graph representations of the dorsoventral (DV) and mediolateral (ML) signaling profiles. In Figure 5B, we quantified the expression domains of *ptc2* and *sox2* to determine the relative timing of the loss of Hh responsiveness versus the loss of neural progenitor identity upon Notch inhibition. In Figure 8D, we quantified the expression domains of *olig2* to determine whether ectopic Gli expression rescues Hh response in Notch^off^ spinal cords.

- Address the important point that Notch inhibition may simply drive cells to differentiate, resulting in the loss of Gli expression and thus loss of competence for a Hh-dependent transcriptional response;

We have performed two different experiments to address this concern. First, to establish the relationship between neural differentiation and Hh responsiveness, we compared the timing of loss of Hh response (*ptc2*) versus loss of neural progenitor state (*sox2*) upon the inhibition of Notch signaling. Our timecourse experiments showed that the loss of *ptc2* expression significantly precedes the loss of *sox2* domain, suggesting that Notch inhibition results in an immediate loss of Hh response, which in turn leads to a loss of neural progenitor state. This new result is shown in subsection “Notch signalling maintains Hh response” and Figure 5.

Second, we performed the experiment suggested by reviewer 1. Briefly, *hsp:EGFP-Gli1* and wild type control embryos were incubated in either DMSO or LY-411575 from 20 hpf to 30 hpf, during which embryos were heat shocked for 30 minutes at 21 hpf (1 hour post drug treatment) to induce EGFP-Gli1 expression. Whole mount in situ hybridization was then performed for *olig2* at 30 hpf. Similar to our experiments shown in Figure 8D-E (induction of EGFP-Gli1 followed by the LY-411575 treatment), ectopic induction of EGFP-Gli1 after 1 hour of LY-411575 treatment was also sufficient to partially restore *olig2* expression. This new result is shown in Author response image 1.

**Author response image 1. respfig1:** Notch signalling regulates Hh response at the Gli level. (**A**) Experimental design. Embryos were heat shocked (HS) for 30 minutes in the solution containing the drug at 1 hour after the drug treatment. (**B**) *hsp:EGFP-Gli1* and wild type control embryos were incubated in either DMSO or LY-411575 from 20 hpf to 30 hpf, during which embryos were heat shocked at 21 hpf. Whole mount in situ hybridisation was performed for *olig2* at 30 hpf. Brackets denote the extent of the spinal cord and lateral views are shown. Arrowheads indicate *olig2* expression in the spinal cord. The *n* number for each staining is shown. Scale bar: 20 μm.

Together, our new data support the model that the competence of neural progenitor cells to respond to Hh signals is actively controlled by Notch signaling, and the loss of Hh responsiveness after Notch inhibition is not an indirect consequence of neuronal differentiation.

- Discuss discrepancies with previously published results in the literature.

We have provided additional discussions to compare and contrast our findings in the context of previously published results. This revised section can be found in subsection “Notch signalling gates Hh responsiveness at the level of Gli transcription factor”.

Reviewer #1:This manuscript addresses the long-standing problem of spinal cord patterning and a recently appreciated, but understudied question of how Notch signaling regulates neural cell responsiveness to Hedgehog signaling.The main strengths of the paper are:1) The manuscript is very nicely organized, written and illustrated. It is very easy to read and understand.2)The work addresses an important problem in neural development and presents new information to the field. Although we have known for a very long time that Notch signaling has a fundamentally important role in maintaining spinal cord progenitors, we still have very little understanding of the details. Recent publications, including from the senior author of this work, have connected Notch signaling to Hh signaling sensitivity. The conclusions drawn by these authors differ substantially from the conclusions of Kong et al.,2015 and Stasiulewicz et al., 2015 and, if valid, would motivate further examination of the mechanistic intersection of Notch and Hh signaling.3) With the exceptions noted below, the data sufficiently support the conclusions.

We thank the reviewer for thoughtful comments on the manuscript and supportive remarks.

The main weaknesses of the paper are:1) There is no indication of the number of trials performed or embryos analyzed for any experiment. Although most of these manipulations probably produce very clear changes in gene expression and having some sort of statistical data might not be so helpful, I do think it does become important for the experiments portrayed in Figure 7D. These experiments are key to the authors' conclusions and it is essential to know if the images they show represent the data accurately.

As mentioned above, we have provided *n* numbers for all experiments throughout the manuscript in figures and figure legends. We have also provided additional quantifications whenever necessary. For PHRESH analysis in Figure 2 and Figure 2—figure supplement 1, we provided graph representations of the dorsoventral (DV) and mediolateral (ML) signaling profiles. In Figure 5B, we quantified the expression domains of *ptc2* and *sox2* to determine the relative timing of the loss of Hh responsiveness versus the loss of neural progenitor identity upon Notch inhibition. In Figure 8D, we quantified the expression domains of *olig2* to determine whether ectopic Gli expression rescues Hh response in Notch^off^ spinal cords.

2) If Notch signaling sensitizes cells to Hh signaling cell autonomously, as expected if Notch does this by acting on Gli1, then a prediction is that the same cells will express ptc2 and her12. However, the double labeling data of Figure 2—figure supplement 3 do not strongly support that prediction. Do higher magnification, higher resolution images reveal that all, or most, ptc2+ cells also express her12?

We have repeated the double fluorescent in situ hybridization and provided new transverse images in Figure 2—figure supplement 3A. Consistent with our model that the same neural progenitor cells express both *her12* and *ptc2*, our results showed that most *ptc2^+^* cells also express *her12*.

3) The most important weakness concerns the series of experiments to investigate where Notch acts in the Hh signaling pathway. The authors' approach is to pharmacologically inhibit Notch signaling at 20 hpf and simultaneously manipulate Hh signaling to determine if a particular manipulation can rescue the patterning deficit resulting from loss of Notch function. The problem is that Notch inhibition causes neural progenitors to rather immediately differentiate. Consequently, any subsequent failure in Hh responsiveness might not be due to absence of Notch activity, but because differentiated cells simply are unresponsive to Hh for unrelated reasons. In fact, an interpretation of Figure 2 is that differentiated cells are not Hh signaling active. This is why the data of Figure 7B are so important and therefore must be rock solid. The implication of these results is that elevation of Gli1 can overcome Notch inhibition to maintain some level of Hh signaling and spinal cord patterning. It appears to me that, in this instance, Gli1 overexpression maintains some neural progenitors and prevents their differentiation despite loss of Notch signaling. I therefore predict that delaying heatshock to induce Gli1 expression by an hour would not have the same effect. If that's the case, I'm not convinced that the data can sufficiently support the Notch effect via Gli1 model.

We thank the reviewer for this excellent point. We have performed two different experiments to address this concern. First, to establish the relationship between neural differentiation and Hh responsiveness, we compared the timing of loss of Hh response (*ptc2*) versus loss of neural progenitor state (*sox2*) upon the inhibition of Notch signaling. Our timecourse experiments showed that the loss of *ptc2* expression significantly precedes the loss of *sox2* domain, suggesting that Notch inhibition results in an immediate loss of Hh response, which in turn leads to a loss of neural progenitor state. This new result is shown in subsection “Notch signalling maintains Hh response” and Figure 5.

Second, we performed the experiment suggested by the reviewer. Briefly, *hsp:EGFP-Gli1* and wild type control embryos were incubated in either DMSO or LY-411575 from 20 hpf to 30 hpf, during which embryos were heat shocked for 30 minutes at 21 hpf (1 hour post drug treatment) to induce EGFP-Gli1 expression. Whole mount in situ hybridization was then performed for *olig2* at 30 hpf. Similar to our experiments shown in Figure 8D-E (induction of EGFP-Gli1 followed by the LY-411575 treatment), ectopic induction of EGFP-Gli1 after 1 hour of LY-411575 treatment was also sufficient to partially restore *olig2* expression. This new result is shown in Author response image 1.

Together, our new data support the model that the competence of neural progenitor cells to respond to Hh signals is actively controlled by Notch signaling, and the loss of Hh responsiveness after Notch inhibition is not an indirect consequence of neuronal differentiation.

Reviewer #2:The manuscript entitled "Notch signalling maintains Hedgehog responsiveness via a Gli-dependent mechanism during zebrafish spinal cord patterning" by Jacobs and Huang investigates the relationship between the Notch (N) and Hedgehog (Hh) pathways in patterning the zebrafish neural tube. Specifically, the authors a technique, PHotoconvertible REporter of Signalling History, as well as in situ hybridization, to demonstrate that the N and Hh pathways are active at the same developmental time and in a common population of cells during spinal cord development. The authors use a combination of small molecule inhibitors, genetic mutants, and transgenic (gain-of-function) zebrafish to show that in the spinal neural tube (but generally not in non-neural tissues), Notch pathway activity is required for Hedgehog responses, while Hh pathway is dispensable for N activity. The authors show that Notch pathway activity is required for Hh signaling at a step downstream of Smoothened and cilia. They show that expression of all of the gli transcription factors (gli1, gli2a, gli2b and gli3) relies on N activity and that forced expression of Gli1 is sufficient to partially restore Hh responses when the N pathway has been inhibited. The fact that the rescue is not complete raises the possibility that the Notch pathway may also serve a permissive role for Gli activity at the post-transcriptional levels.This is important work that advances our understanding of the Hh/N relationship. While previous findings have pointed to a functional relationship, the current work significantly advances our understanding of the mechanistic nature of the relationship and suggests the mechanism acting in zebrafish may differ from that acting in the mouse (Stasiulewicz et al., 2015). The work has been carefully conducted, although it could benefit from some quantitative analysis. Specific comments that relate to overall flow of the writing, data analysis, and interpretation of results are listed below.

We thank the reviewer for a critical reading of the manuscript and the constructive suggestions.

Specific comments:1) To clearly emphasize the importance of the work, it would be useful to emphasize how this work addresses a previously unanswered question in the introduction section. While it is important, with respect to cell fate specification, that the competence of neural cells to respond to Hh signals be shut down at some point in later development, the mechanism by which this competence is restricted is not understood. By demonstrating that N activity is also time-limited and corresponds to the stages when cells are competent for Hh responses) and that Hh responses are dependent on N activity, the authors provide a clue as to the nature of this temporal gating of Hh responsiveness.

We thank the reviewer for this excellent suggestion. We have revised the introduction to highlight the importance of Hh signaling termination in cell fate specification and also emphasize how the temporal gating of Hh responsiveness is controlled is still poorly understood.

2) N values should be clearly stated for each experiment and quantification of data in some of the experiments should be performed.

As mentioned above, we have provided *n* numbers for all experiments throughout the manuscript in figures and figure legends. We have also provided additional quantifications whenever necessary. For PHRESH analysis in Figure 2 and Figure 2—figure supplement 1, we provided graph representations of the dorsoventral (DV) and mediolateral (ML) signaling profiles. In Figure 5B, we quantified the expression domains of *ptc2* and *sox2* to determine the relative timing of the loss of Hh responsiveness versus the loss of neural progenitor identity upon Notch inhibition. In Figure 8D, we quantified the expression domains of *olig2* to determine whether ectopic Gli expression rescues Hh response in Notch^off^ spinal cords.

3) As gli2a, gli2b and gli3 expression in the spinal cord is largely eliminated in Notch(off) spinal cord, is it possible that the failure of hsp:EGFP-Gli1 to fully restore the Olig2+ motor neuron precursor domain is not due to the role of Notch in maintaining Gli1 activity at the post-transcriptional level but, rather hsp:EGFP-Gli1 expression may not be able to fully specify the Olig2 fate in the absence of additional activity from gli2a, gli2b and gli3. This possibility should be discussed.

We thank the viewer for pointing out this alternative scenario. We have revised the text to discuss this possibility in subsection “Ectopic expression of Gli1 partially rescues Hh response in Notchoff spinal cords” and in subsection “Notch signalling gates Hh responsiveness at the level of Gli transcription factors”.

Reviewer #3:The paper by Jacobs and Huang tackles a very interesting problem in the field of signal interpretation. They identify a tissue-specific regulation of Hh signaling that is both surprising and intriguing. They use a range of mutant embryos to uncover the mechanism of this regulation, and identify that this regulation is at the level of Gli itself.The paper is logically presented and the argument r.e. Hh regulation, constructed sensibly.

We thank the viewer for a critical reading of our manuscript and the supportive comments.

However, I have a major concern regarding the imaging and analysis. There is very little quantification of data. We are shown snapshots of "representative" embryos at different timepoints under different conditions, and we are encouraged to take the authors word on the results. For the approach outlined, the authors should be quantifying signal levels and recording how they change in time across multiple embryos. What is the embryo-to-embryo variability? How spatially confined are the results? What is the cell-to-cell variability? The quality of the presented fluorescent images is poor and somewhat unconvincing. There is a lot of information in the PHRESH reporter that they are missing because of out-dated analysis approaches.

As discussed above, we have provided *n* numbers for all experiments throughout the manuscript in figures and figure legends. We have also provided more in-depth quantifications for the key experiments (Figure 5B and Figure 8D). For PHRESH analysis, transverse views were obtained by 3D reconstruction from confocal stacks imaged from lateral views of the spinal cord. Since the axial resolution in the z-dimension is typically less than the lateral resolution in the x- and y-dimension in confocal microscopy, the reconstructed images were of slightly lower resolution compared to lateral images. Despite this limitation, the reconstructed transverse views clearly showed the spatial and temporal dynamics of Notch and Hh signaling response in the spinal cord during embryo development. To further compare the signaling dynamics, we quantified Kaede^green^ fluorescence intensity (active signaling) along the dorsoventral (DV) and mediolateral (ML) axes to generate signaling profiles. New graphs are shown in Figure 2 and Figure 2—figure supplement 1. In particular, quantifications of signaling profiles across multiple embryos showed consistent trends with small variability (Figure 2—figure supplement 1). Comparison of DV/ML signaling profiles across different stages (Figure 2 and Figure 2—figure supplement 1) supports our general conclusions that Notch and Hh signaling display similar spatiotemporal kinetics during spinal cord patterning, going through signaling “activation”, “consolidation” and “termination” phases (subsection “Notch and Hh signalling display similar dynamics during spinal cord patterning”). In this manuscript, we focus on comparing signaling profiles of the entire spinal cord. However, we have previously shown that PHRESH analysis can be combined with antibody staining to examine the temporal dynamics of Hh response at the single cell resolution (Huang et al., 2012).

A second major issue I have regards the means of perturbation. They rely entirely on drug perturbations to Notch. Are heat shock or Gal4-driven dominant negatives alleles available to test the suppression? The problem with drug treatments is that the whole embryo is affected so deciphering tissue-specific phenotypes needs to be done carefully. Both drug and heat-shock induced perturbations should be used to cross-validate the approach.

We have used different tools to examine the effect of Notch inhibition. First, we utilized the *mindbomb* mutant, which has compromised Notch signaling due to defects in the endocytosis of Notch ligands (Figure 3—figure supplement 1A). Second, we injected wild-type embryos with morpholinos targeting both *rbpja* and *rbpjb* genes (previously known as *Su(H)1* and *Su(H)2*), which encode DNA-binding transcription factors required for Notch response (Figure 3—figure supplement 1B, new data). In both *mindbomb* mutants and *rbpja/b^MO^*-injected embryos, Hh response was completely lost in the spinal cord but remained largely normal in the somites. This phenotype was recapitulated by LY-411575, a specific inhibitor of Notch signaling (Figure 3). The loss of Hh response in the spinal cord was similarly observed when Notch signaling was blocked by compound E, a different small molecule inhibitor of Notch signaling (Huang et al., 2012). Even though the whole embryo is affected in both drug treatments and genetic mutants, the fact that Notch inhibition results in distinct phenotypes on Hh response in different tissues (the spinal cord versus the somites) in the same embryos is a strong indication that Notch signaling regulates Hh responsiveness in a tissue-specific manner. As drug inhibition provides precise temporal control with fast kinetics of signaling inhibition (Figure 1—figure supplement 2), this was our preferred method of Notch inhibition. As suggested by the reviewer, we also obtained a construct containing a dominant negative form of murine Mastermind-like (MAML) gene tagged with GFP under the control of the heat shock promoter (*hsp:dnMAML-GFP*) (Zhao et al., 2014). MAML is a transcriptional co-activator required for Notch-mediated transcription. Unfortunately, embryos injected with *hsp:dnMAML-GFP* showed only partial inhibition of Notch signaling upon heat shock induction, and it was therefore inconclusive to examine its effect on Hh response.

To summarise, I think the paper is potentially very interesting and tackles a problem relevant to a broad range of researchers. However, in its current form it appears dated and does not use suitable analysis techniques that give me sufficient confidence in the results.

We thank the reviewer for recognizing the importance of our work. As discussed above, we have provided *n* numbers for all experiments throughout the manuscript in figures and figure legends. We have added in-depth quantifications on the PHRESH analysis as well as several key experiments. The revised and newly added figures are Figure 2, Figure 5, Figure 8D, Figure 2—figure supplement 1, Figure 2—figure supplement 3A, and Figure 3—figure supplement 1B.

Additional references:

Zhao L, Borikova AL, Ben-Yair R, Guner-Ataman B, MacRae CA, Lee RT, Burns CG,

Burns CE. 2014. Notch signaling regulates cardiomyocyte proliferation during

zebrafish heart regeneration. Proc Natl Acad Sci U S A 111:1403–8.

doi:10.1073/pnas.1311705111